# Long-term erosion of the Nepal Himalayas by bedrock landsliding: the role of monsoons, earthquakes and giant landslides.

Odin Marc[1], Robert Behling[2], Christoff Andermann[2], Jens M. Turowski[2], Luc Illien[2,3], Sigrid Roessner[2], and Niels Hovius[2].

[1]École et Observatoire des Sciences de la Terre - Institut de Physique du Globe de Strasbourg, Centre National de la Recherche Scientifique UMR 7516, University of Strasbourg, 67084 Strasbourg Cedex, France.
[2]Helmholtz Centre Potsdam, German Research Center for Geosciences (GFZ), Telegrafenberg, 14473 Potsdam, Germany
[3]Laboratoire de Géologie, Ecole Normale Supérieure, 24 Rue Lhomond, 75000, Paris, France.

*Correspondence to*: Odin Marc (odin.marc@unistra.fr)

**Abstract.**

In active mountain belts with steep terrain bedrock landsliding is a major erosional agent. In the Himalayas, landsliding is driven by annual hydro-meteorological forcing due to the summer monsoon and by rarer, exceptional events, such as earthquakes. Independent methods yield erosion rate estimates that appear to increase with sampling time, suggesting that rare, high magnitude erosion events dominate the erosional budget. Nevertheless, until now, neither the contribution of monsoon and earthquakes to landslide erosion, nor the proportion of erosion due to rare, giant landslides have been quantified in the Himalayas. We address these challenges by combining and analyzing earthquake and monsoon induced landslide inventories across different timescales. With time-series of 5 m satellite images over four main valleys in Central Nepal, we comprehensively mapped landslides caused by the monsoon from 2010 to 2018. We found no clear correlation between monsoon properties and landsliding, and a similar mean landsliding rate for all valleys, except in 2015, where the valleys affected by the earthquake featured ~5-8 times more landsliding than the pre-earthquake mean rate. The long-term size-frequency distribution of monsoon induced landslides (MIL) was derived from these inventories and from an inventory of landslides larger than ~0.1 $km^2$ that occurred between 1972 and 2014. Using a published landslide inventory for the Gorkha 2015 earthquake, we derive the size-frequency distribution for earthquake-induced landslides (EQIL). These two distributions are dominated by infrequent, large and giant landslides, but underpredict an estimated Holocene frequency of giant landslides (>1 $km^3$) which we derived from a literature compilation. This discrepancy can be resolved when modelling the effect of a full distribution of earthquakes of variable magnitude and considering that shallower earthquakes may cause larger landslides. In this case, EQIL and MIL contribute about equally to a total long-term erosion of ~2 +/-0.75 $mm.yr^{-1}$ in agreement with most thermochronological data. Independently of the specific total and relative erosion rates, the heavy-tailed size-frequency distribution from MIL and EQIL and the very large maximal landslide size in the Himalayas indicate that mean landslide erosion rates increase with sampling time, as has been observed for independent erosion estimates. Further, we find that the sampling time scale required for adequately capturing the frequency of the largest landslides, which is necessary for deriving long-term mean erosion rates, is often much longer than the averaging time of cosmogenic [10]Be methods. This observation

presents a strong caveat when interpreting spatial or temporal variability of erosion rates from this method. Thus, in areas where very large, rare landslide contributes heavily to long-term erosion (as the Himalayas), we recommend [10]Be sample in catchments with source areas >10,000 km², to reduce the method mean bias below ~20% of the long-term erosion.

## 1 Introduction

In some locations erosion rates appear to increase with measurement time. A possible explanation is that rare, catastrophic erosion events dominate the long-term erosional budget (Kirchner et al., 2001). This explanation implies that a full understanding of sediment fluxes and landscape dynamics, and their relations to tectonic and climatic forcing, can only be realized with erosion estimates covering long timescales while any short-term measurements are not representative of these dynamics. To test and quantify this hypothesis it is necessary to

constrain both the erosion associated with continuous, unexceptional forcing and with extreme forcing events. In the Nepal Himalayas many studies have characterized erosion rates over different time scales. Short-term (1-10 yr) average erosion rates based on fluvial sediment measurements in Nepal, vary between 0.1 and 2 mm.yr$^{-1}$ for small (100-3000 km²) catchments (Gabet et al., 2008), but are typically as high as 1-2 mm.yr$^{-1}$ for principal catchments draining the mountain belt (Andermann et al., 2012, Morin et al., 2018, Struck et al., 2016). Catchment-wide mean

erosion rates derived from [10]Be concentrations in river sediment from across the Himalayas typically yield erosion rates of 0.5-2 mm.yr$^{-1}$ (Vance et al., 2003, Godard et al., 2012, 2014, Scherler et al., 2014, Portenga et al., 2015, Abrahami et al., 2016), averaged over ~300-1200 years. Uncertainty remains substantial given that each study reports a number of outliers (<0.1 or >2 mm.yr$^{-1}$), possibly due to recent landsliding or incomplete mixing. On geological timescales (0-2 Myr), fission track data inverted with thermomechanical models indicates exhumation

rates of 2-3 mm.yr$^{-1}$ in the High Himalayas of central Nepal (Wobus et al., 2005, 2006, Hermann et al., 2010, Thiede and Ehlers, 2013), possibly up to 5 mm.yr$^{-1}$ (Burbank et al., 2003, Whipp et al., 2007). This ensemble entails an increase of erosion rates with increasing measurement timescales, as well as a high spatial variability of erosion rates at short and intermediated timescales. Although well established, the origin of these features is poorly

understood and may be attributed to an inadequate average of extreme events over short timescales, even if climatic variations since the Pleistocene may also have modulated erosion.

In steep terrain, which is prevalent throughout the Himalayas, mass-wasting is considered the dominant erosional process on hillslopes and the main source of sediment to rivers (Burbank et al., 1996, Hovius et al., 1997, Hovius et al., 2000, Gabet et al., 2004, Morin et al., 2018, Struck, et al., 2015). Most landslides are triggered by elevated pore-pressure due to heavy rainfall or snowmelt (Van Asch et al., 1999, Iverson 2000) or by ground shaking caused by shallow earthquakes (Keefer et al., 1984, Marc et al., 2016a, Tanyas et al., 2017). Tracking pore pressure at the landslide scale is difficult, but studies of landslides or landslide populations triggered by rainfall have reported a non-linear, often power-law, increase of the landslide density or total area or volume with rainfall metrics such as intensity, duration, and especially total rainfall (Burtin et al., 2013, Chen et al., 2013, Saito et al., 2014, Marc et al., 2018). For earthquakes, a linear scaling of landslide density with peak ground acceleration beyond a threshold acceleration is consistent with the spatial pattern and total area and volume of landslide populations caused by earthquakes (Meunier et al., 2007, 2013, Marc et al., 2016a, 2017). Temporal coincidence of these two independent forcings enhances landsliding, and it has been shown that landslide susceptibility to rainfall is elevated in the epicentral zone of large, shallow earthquakes, followed by a progressive decay to pre-seismic values (Marc et al., 2015). Thresholds and non-linear scaling reported in various studies imply that long-term erosion is influenced by the frequency-intensity distribution of the triggering events (seismic or meteorologic) associated with a given climatic and tectonic setting (e.g., Marc et al., 2016b). In turn, the landslide size distribution can be characterized by power-law behaviour beyond a cut off size, and is often heavy-tailed when converted to volume (c.f. Hovius et al., 1997, Stark and Hovius 2001, Malamud et al., 2004). This implies a disproportionate role of rare, large events in setting long-term erosion rates. The roll-over and divergence from power law behaviour

has been interpreted as an effect of resolution censoring (Stark and Hovius 2001) or as emerging for mechanical reasons (Stark and Guzzetti 2009, Frattini and Crosta 2013, Milledge et al., 2014).

Independent of the trigger, landslide occurrence may be due, to an extent, to an increased propensity to slope failure due to rock mass weakening and the development of discontinuities, for example due to weathering, mineralization, mechanical fatigue (cf. Lacroix and Amitrano 2013, Riva et al., 2018). However, here we will not focus on these aspects, since systematically monitoring and quantifying these predisposing factors remains challenging. Instead, we aim to quantify the long-term landslide erosion caused by earthquake and monsoon occurrence, and its dependence on rare and large landslides. It is generally accepted that in the Himalayas, widespread landsliding is driven by the annual summer monsoon (Monsoon-Induced Landsliding, MIL) (e.g. Gabet et al., 2004, Andermann et al., 2012, Struck et al., 2015), with its prolonged intense rainfall, and by less frequent high magnitude forcing events, such as earthquakes (Schwanghart et al., 2016, Stolle et al., 2017, Roback et al., 2018). However, until now the influence of monsoon properties on annual landsliding has remained poorly constrained, in part because comprehensive landslide mapping is limited (e.g., Dahal and Hasegawa, 2008). In contrast, the intense effort of landslide mapping throughout Nepal following the 2015 Gorkha earthquake allows for the first time an estimate of the contribution of earthquake-induced landsliding (EQIL) to long-term erosion in the Nepal Himalayas. Mapping of the landslides due to monsoon rainfall following the earthquake offers an opportunity to constrain the seismic perturbation of the landscape. Finally, to assess if rare, giant landslides (>km³) contribute significantly to erosion and can explain the discrepancy between short and long-term erosion (Weidinger, 2011, Zech et al., 2009), it is necessary to constrain the size-frequency distribution of landslides associated with the different triggers. In the Himalayas, glaciers do not seem to contribute much to the erosion budget of the range (Morin et al., 2018), likely because in spite of having significant local effects on the erosion dynamics (e.g., Heimsath and McGlyyn, 2008) they have a very limited areal extent, even during ice ages. Thus,

we consider that quantitative understanding of role and behavior of landsliding in the Himalayas can be obtained without investigating glacial and peri-glacial areas.

Here we use several multi-temporal landslide inventories from the High Himalayas of Nepal to constrain the erosion associated with recent monsoons and the Gorkha earthquake and its aftermath. With a 50 year record of large landslides and an estimate of earthquake recurrence time, we constrain the size-frequency distribution of both MIL and EQIL. We show that it is consistent with a ~10,000 year record of dated giant landslide deposits, constraining the maximum landslide size and allowing quantification of long-term landslide erosion due to tectonic

and climatic forcing. We find that landslide erosion is dominated by the largest landslides and that, when integrated over the relevant size or frequency range, it matches independent erosion rate estimates obtained over various timescales (yr, kyr, Myr). Hence, the size and recurrence time of the largest landslides in a mountain belt has important implications for the interpretation of erosion patterns derived from techniques averaging over short (e.g., fluvial sediment budget) to intermediate (e.g., [10]Be) timescales.


**1 Data and Methods**

**2.1 Landslide inventories: satellite imagery, landslide mapping and dated deposit compilation**

We mapped landslides triggered during eight monsoon seasons (2010-2017), and by the Gorkha earthquake (25 April 2015) and its largest aftershock (12 May 2015) using a series of 5m-resolution Rapid Eye (RE) images (Suppl.

Table 1, Fig. 1). We focus on four study areas, delimited by Rapid Eye (RE) satellite image tiles (4552225, 4552106, 4552007 and 4551910), each ~25 by 25 km, and together representing 2300 km² of mapped area, as well as 210 km$^2$ of (peri-) glacial terrain where the absence of vegetation did not allow mapping. Indeed, the change from a vegetation signature to a rock debris signature is very conspicuous in multispectral imagery, even for sparse vegetation, whereas textural or spectral changes in rocky/sedimentary surfaces remain challenging to detect and

interpret. We chose the four tiles to cover the High Himalayan section with steep relief and focused erosion. One RE tile, covering a part of the Kali Gandaki catchments (KG), lies outside of the area affected by the 2015 Mw 7.8 Gorkha earthquake and is used as a benchmark for non-seismic erosion rates. The three other tiles, located over the Buri Gandaki (BG), Trisuli (T) and Bhote Koshi (BK) catchments cover representative sections of the rupture zone of the Gorkha earthquake. The BK area is also less than 20km away from the epicentre of the Mw 7.3 aftershock of 12 May 2015 that was reported to have triggered additional failures in this area (Fig. 1). We used the map of coseismic landslides by Roback et al. (2018) and refined the mapping in the BK area, where available imagery allowed differentiation between failures due to the Gorkha earthquake and the large aftershock.

To obtain our landslide maps, we used, in a first step, a landslide mapping algorithm (Behling et al., 2014, 2016) applied to time series optical remote sensing data. The approach comprises automated pre-processing routines (e.g. geometric co-registration, masking of clouds, water and snow) and multi-temporal change detection methods, resulting in potential landslide objects, which are assigned a probability of actually being a landslide. The change detection builds on the analysis of temporal NDVI-trajectories, representing footprints of vegetation cover changes over time. Limited amount of imagery did not allow for accounting for and removing seasonal variations in the NDVI signatures, but most of the scenes are in the post-monsoon season when vegetation cover is highest, limiting such variations (Table S1). Landslide-specific trajectories are characterized by short-term destruction of vegetation cover and longer-term revegetation resulting from landslide related disturbance and dislocation of fertile soil cover. The consideration of this multi-temporal vegetation signal further minimizes the effect of possible seasonal variations. In combination with slope gradient and parallelism to rivers, which enhance the exclusion of anthropogenic (building, field clearings) and flood related disturbance, respectively, this approach enables automated identification of landslides of different sizes and shapes and in different stages of development (e.g. fresh occurrences and reactivations of existing landslides) under varying natural conditions.

The output of the algorithm was visually inspected and necessary corrections were applied manually. A specific concern was the adequate splitting and re-dating of multiple adjacent landslides bundled into single polygons by the algorithm. In our case, the splitting of amalgamated polygons is not only important for correct volume estimates (Marc and Hovius, 2015), but also for attribution of each polygon to the appropriate triggering period. Manual splitting, or remapping when needed, were based on inspection and comparison of the multispectral imagery and on the topographic context. Another important step was the removal of erroneously detected landslides, for example debris and clearings related to road construction or to fields near villages. Then, polygons related to debris flows and/or significant fluvial channel disturbance were reduced to their source and runout areas upslope of channels with permanent discharge, as visible in the RE imagery. Thus, mapping of debris flow areas and their erosional impact is limited to hillslopes and excludes areas of alluviation or flooding mostly affected by depositional processes. Nevertheless, the volume of such debris flows is difficult to estimate based on our mapping information (cf. 2.2). Last, in the Trisuli RE tile, we noticed through visual inspection at least four large (0.1 to 0.4 km$^2$) hillslope segments that had downslope displacements of several meters in some years, but seemed immobile in others. We do not include these mobile hillslope segments in our analysis as they did not yet practically fail, but they may contribute to the sediment export from this catchment in the future. Potential links between annual movements and the monsoon rainfall are unclear and further investigation would require proper quantification of the block movement history, out of the scope of this work.

The selected areas and time periods covered by RE imagery may not be large enough to robustly constrain the mean frequency of very large and rare landslides. To obtain a regional handle on the occurrence of such landslides, we compared a series of cloud free Landsat images (Suppl. Table 2), covering an area of 11,750 km$^2$ in central Nepal (after excluding ~ 3700 km² of (peri-) glacial areas where reliable mapping was not possible). The four RE tiles are located within this larger High Himalayan region, which stretches ~315 km long and ~48 km wide from Dhaulagiri to the Bhote Koshi valley (Fig. 1). This area encompasses several lithological units, a climatic

gradient (with enhanced precipitation south of the high peaks, and a rain shadow behind), localized glaciated areas and a likely uplift gradient (Fig S1 or 1). However, the overall result of these heterogeneities on landsliding is unclear and we start by assuming subparts of our study area (e.g., RE tiles, region of coseismic landsliding) have a similar behaviour and can be compared applying only an areal normalization, and will discuss the validity and caveats of this assumption at the end. Within the larger region, we mapped all new landslides larger than ~0.08 km² between 1972 and 2014 (Fig. 1). A direct comparison of the newest and oldest images (2014 and 70-80s) did not allow detection of all failures because of partial revegetation, occasional shadows or successive phases of failure at the same site. Therefore we combined imagery obtained approximately every decade from 1972 to 2014, to have a full coverage of the area of interest with a very low proportion (<5%) of areas obscured by cloud or topographic shadows (Supplementary Table 2). We note that, with the exception of the 12 landslides mapped on the last images and two obscured in the first images taken after their occurrence, we could constrain revegetation rates (i.e., the time required for vegetation to recolonize most of the scarp and make it indistinguishable from the surroundings in the available imagery) for the 35 remaining large landslides in our data set. Only ten of these were not distinguishable on the second image after their occurrence, meaning that they had fully revegetated in less than about 12 years. The other 25 (70%) had revegetation times longer than 11 years and longer than 20 years in 11 cases. It is thus unlikely that a substantial number of large landslides could have remained undetected because they occurred and revegetated between two mapping frames. Therefore, we consider that the inventory is representative of the mean frequency of large landslides over the 4 last decades.

The last dataset we use is a literature compilation of giant landslides deposits, with volumes typically >1 km³, that can be used to constrain the age and size of the largest landslide events in the Himalayas (Fig. 1). The Tsergo Ri (Langtang) and Braga (Manang) landslides are the largest reported events, with estimated volumes of 10-15 km³ (Weidinger et al., 2002, Weidinger, 2006, Fort, 2011). However, these two landslides have been

significantly eroded during the last glacial period and it is unclear if the imprint of other landslides has been reliably preserved. Nevertheless, they are good examples of single giant landslides, one a peak collapse (Tsergo Li) and the other the collapse of the northern flank of the Annapurna (Braga), and they can be used to constrain the likely maximum landslide size and a minimum probability of occurrence since the last glacial. A more complete picture exists for absolute or relative dating of very large landslide deposits of Holocene age, along the portion of the range covered by our Landsat inventory. We found reference to deposits of three giant landslides around the Annapurna range dated to within the last ~5000 years, the Dhumpu (Upper Kali Gandaki) (~3 km³), Latamrang (Marsyangdi) (~5 km$^3$) and Sabche (Pokhara) (~4-5 km$^3$) landslides, respectively (Fort, 2011, Zech et al., 2009, Pratt-Sitaula et al., 2004, Schwangart et al., 2016). To these we add the Dhikur (Marsyangdi) landslide (~1 km$^3$), which is considered post-glacial in the absence of an absolute date (Weidinger, 2006, Fort, 2011). The 6 deposits mentioned above represent a complete list of giant landslides (>1 km$^3$) present in our area and discussed in the literature (Table 1), and in a twice longer swath (from Dolpo to Sikkim), only three other deposits >1km³ are known and attributed to giant landslides, the Ringmo, Khumjung and Dzongri deposits, which are all considered to be interglacial (Fort, 2011, Weidinger and Korup, 2009). Other massive terrace deposits in valleys in the High Himalayas result from catastrophic sedimentary events (e.g., Cenderelli and Wohl, 1998, Pratt-Sitaula et al., 2007, Lave et al., 2017), but their conditions of formations are diverse (Glacial lake outburst floods, multiple debris flow, giant landslide evacuation) and relating them to individual landslides challenging. Importantly, to accurately estimate the frequency of a given landslide size, deposits should be attributable to single landslides and not result from cumulative deposition. Geomorphological and petrographic evidence suggests single failures for all events in our catalogue (Weidinger et al., 2002, Weidinger, 2006, Fort, 2011), except for the Sabche landslide, where dating and morphology of the sediment suggest three major deposition events over three hundred years (Schwanghart et al., 2016). This case could be a major, single landslide with prolonged debris flow transport, or correspond to three sub-events with an average volume of ~1.5 km$^3$. Based on our literature survey, we consider that at least four giant

landslides (1-5 km$^3$) occurred in our study region during the Holocene, although the deposits may originate from up to six giant failures. The actual upper limit of giant landslide frequency is hard to constrain given that in spite of their size and impact on the landscape, their deposits are not always recognizable from remote sensed imagery (Weidinger and Korup, 2009), and remote valleys that are less well investigated may still hold some undiscovered
deposits.

### 2.2 Volume estimation and runout correction

Landslide plan view area, A, and perimeter, P, were directly obtained from each mapped polygon. These values represent the total area disturbed by a landslide, including the scar, runout and deposit areas, because a systematic
delineation of the scar was not possible from most of the available imagery. Hence, estimates of landslide volume, which are based on area, may be excessive slides with long runout. We applied a correction for runout proposed by Marc et al. (2018), allowing estimation of the landslide width, scar area and volume. First, assuming that each landslide has an elliptical shape, its mean width, W, is computed based on P and A. With 418 landslide polygons, mapped from medium (10 to 30 m) and high resolution (1 m) imagery, they found that for 72% and 96% of the
widths estimated with this method were within 30% to 50%, respectively, of the actual (measured) scar width (Marc et al., 2018). The bias was randomly distributed across a wide range of area ($10^2$-$10^5$ m$^2$), aspect ratio (2-30) and environment (with landslides from Japan, Colombia, Brazil and Taiwan). Second, the scar area is estimated as As = 1.5 W$^2$, using the mean length/width ratio of a worldwide database composed of 277 landslide scars with volumes ranging from 1000 m$^3$ to 1 km$^3$ (Domej et al., 2017). We note that the distribution of estimated landslide scar sizes,
based on our geometric correction of the landslides triggered by the Gorkha earthquake, is similar to the one derived from scar outlines independently mapped from satellite imagery (Roback et al., 2018, Fig. Suppl. 1). However, our estimates of scar area are about 50-100% larger than those of Roback et al. (2018), as their mapping was conservatively limited to the very upper part of the landslides, with a length width ratio often less than 1. Finally,

we converted landslide scar area, As, into volume, V, with the relation $V=\alpha As^{\gamma}$, with parameters for shallow

landslide scars ($\gamma=1.262$ +/- 0.009; $\log10(\alpha)=-0.649$ +/- 0.021) and bedrock landslide scars ($\gamma=1.41$+/- 0.02;

$\log10(\alpha)=-0.63$ +/- 0.06) for $As<10^4\,m^2$ and $As>=10^4\,m^2$, respectively (Larsen et al., 2010). For reference, we also

computed landslide volume with the whole landslide area and using whole landslide parameters ($\gamma=1.332$ +/- 0.005;

$\log10(\alpha)=-0.836$ +/- 0.015) for landslides with $A<10^5\,m^2$, and bedrock landslide parameters ($\gamma=1.35$ +/- 0.01;

$\log10(\alpha)=-0.73$ +/- 0.06) for larger landslides (Larsen et al., 2010). In this study, all analyses of landslide area and

volume are performed after the runout correction, while results without this correction are presented in the

supplementary materials (Fig. Suppl., 2, 3).

Uncertainties in this approach include the 1-sigma variability of the coefficient and exponent of the

landslide area-volume relations given above, and an assumed standard deviation of 20% of the mapped landslide

area (Marc et al., 2016a, 2018). These uncertainties were propagated into the volume estimates assuming a Gaussian

distribution of errors. The standard deviation of the total landslide volume, for entire catalogues or for local subsets,

were calculated assuming that the volume of each individual landslide is unrelated to that of any other in the data

set, thus, ignoring possible co-variance. Although estimated 2-sigma for single landslides is typically from 60 to

100% of the individual volume, the 2-sigma uncertainty for the total volume of inventories with 100-1000

landslides is typically below 10-20% (Marc et al., 2016a, 2018).


### 2.3 Spatio-temporal frequency of landsliding for the estimation of long-term erosion rates

Long-term erosion rates can be derived by integrating the spatio-temporal frequency ($yr^{-1}.km^{-2}$) of landslides from

the smallest to the maximum landslide size (Hovius et al., 1997). To estimate landslide size-frequency distributions,

we computed a histogram of landslide area (whole or scar), using log-spaced bins, and then normalized by the

mapped area, $A_{map}$ (cf 2.1), and the timespan during which landslides occurred, $T_{map}$. We computed the size-

frequency distribution for four inventories, the landslides induced by the Gorkha earthquake as mapped by Roback et al., (2018), the 2010-2017 monsoons mapped from RE imagery, the 1972-2014 monsoons mapped from Landsat imagery, and the compilation of giant ($>1$ km$^3$) landslide deposits in central Nepal.

Here, we review $A_{map}$ and the considerations leading to the values of $T_{map}$ for each of the inventories. For the earthquake inventory we use $A_{map}=7000$ km$^2$, that is the area of intense landsliding across the high Himalayas, ignoring sparse landsliding in the lesser Himalayas and the Siwaliks (Martha et al., 2016, Roback et al., 2018). For an earthquake trigger, $T_{map}$ must represent an average earthquake recurrence time. Studies of paleo-ruptures in central Nepal, constrained by historical damage or dated fault scarps, have revealed complex earthquake intervals (Mugnier et al., 2013, Bollinger et al., 2014, 2016). Specifically, data from historical reconstructions, accounting for blind ruptures, suggests that at least six large earthquakes affected central Nepal in the last ~1000 years, possibly eight if we consider ruptures from Eastern and Western Nepal that may have propagated to Central Nepal (Mugnier et al., 2013, Bollinger et al., 2016). However, these ruptures have poorly constrained magnitudes, varying from Mw ~ 7.5 to 8.5, and uncertain return times (Mugnier et al., 2013). Dated deformation of river terraces in the last 4500 years indicates relatively regular surface rupturing of the Main Frontal Thrust (MFT) by great earthquakes every 650-850 yr (Bollinger et al., 2014). If they were similar to the Bihar rupture, the most recent event on the MFT, then the corresponding earthquakes would have had Mw ~ 8.1-8.4 (Bollinger et al., 2014). Hence, we consider a ~750 year return time of great surface rupturing earthquakes of Mw ~8.3 and use a Gutenberg-Richter law with b-value of 1, consistent with instrumental and historical data in Nepal (Avouac, 2015), to estimate a return time of ~300 years for a Mw 7.9 event. The additional contributions to mass wasting by more frequent earthquakes with an intermediate magnitude (i.e. Mw~7) as well as infrequent giant earthquakes (Mw 8.5) are likely to be important, but cannot be constrained from currently available landslide inventories and we will discuss a correction based on modelling results.

For the RE inventory $A_{map}$=2300 km$^2$. The landslide area histogram must be normalized by the number of monsoon years (=8) covered by the imagery. However, if some years are significantly affected by the occurrence of the Gorkha earthquake, then they may not be representative of the monsoon forcing, and should be excluded, reducing $T_{map}$ for this dataset. Below (cf., 3.1.3), we constrain the duration of the influence of this earthquake on rainfall-induced landslide rates.

For the Landsat inventory, we mapped an area $A_{map}$=11750 km$^2$ along the range, using imagery spanning from 1972 to 2014. However, we use $T_{map}$=46 years, to include the 1968 Labubesi landslide (Weidinger, 2011), which is clearly visible in the 1972 imagery. It is the second largest failure of this inventory (0.6 km$^2$). In doing so, there is a possibility that we slightly underestimate the frequency of smaller landslides in this catalogue, but we probably obtain a better average of the larger ones by considering this additional failure and the slightly longer time span.

The compilation of Holocene giant landslide deposits is considered representative of the whole area of interest with $A_{map}$=11750 km$^2$ and $T_{map}$=10,000 years, yielding a range of frequency of ~3 to $6.10^{-8}$ yr$^{-1}$.km$^{-2}$. Assuming a typical volume of ~3 km$^3$, the scar areas of these giant landslides can be back-estimated based on A-V relationships (cf 2.2), to a range of 11 to 26 km$^2$.

To estimate the long-term erosion due to landsliding in the Nepal Himalayas, we convert mapped landslide area to volume (cf 2.2) and numerically integrate the size-frequency relations for landslide scars with surface areas until the maximum scar size, back-estimated as 40 km², from the largest deposit in the area (10-15km³, in Langtang (Weidinger et al., 2002)).

## 3. Results

### 3.1 Landslide inventories and erosion across timescales

#### 3.1.1 Seismically triggered landslides

305        In the RE tile over the Bhote Koshi (BK) we mapped 953 landslides attributed to the Gorkha earthquake and a further 167 due to the large Mw 7.3 aftershock on 12 May 2015. With the runout correction proposed in 2.2 we estimate a total scar area of 1.25 and 0.14 km$^2$ (i.e., a density of 2000 and 230 m$^2$.km$^{-2}$), and a total volume of 3.1 and 0.22 Mm$^3$ (i.e., 5 and 0.35 mm of erosion), respectively. In the Kali Gandaki area (KG), we detected only 5 new landslides in May 2015, which could have been triggered by the earthquake, or by pre-monsoon rainfall in

April of that year. This is consistent with other studies that do not report coseismic landsliding in this area (Martha et al., 2016, Roback et al., 2018). In the Buri Gandaki (BG) and Trisuli (T) areas, about 2400 and 1600 coseismic landslides were reported by Roback et al., (2018), consistent with the new failures visible in the RE imagery, although some landslide outline polygons appear distorted, likely due to orthorectification issues of the imagery they used. After runout correction, we estimate a total scar area of 2.0 and 2.1 km$^2$ (i.e., a density of 4200 and 3300

m$^2$.km$^{-2}$), and a total volume of 8.3 and 11 Mm$^3$ (i.e., 17 and 18 mm of erosion), in the BG and T areas, respectively. Next, we examine how landsliding due to instantaneous seismic forcing compares with the steady landslide flux due to annual monsoons.

#### 3.1.2 Monsoon-driven landsliding

        In the four areas covered by our RE imagery, from west to east KG, BG, T and BK, we mapped a total of

4937 landslides, with a cumulative area of 14.6 km$^2$ in the 8 monsoon seasons between 2010 and 2017.

        The 2015 Gorkha earthquake may have changed the propensity to rainfall-induced slope failure in subsequent years (cf. Marc et al., 2015). Therefore, we limit our initial analysis of monsoon-driven landsliding to the 5 years preceding the earthquake. In this time window, the total area of landslide scars activated by each monsoon, normalized by mapping area, is very similar in the four catchments, ranging from ~50 to 200 m$^2$.km$^{-2}$

with a mean of 133 +/- 57 (+/- are 1-sigma unless specified) $m^2.km^{-2}$ for the four mapping tiles combined (Fig 2).

Landslide volume density and erosion are more scattered, ranging from 100 to 1000 $m^3.km^{-2}$ (i.e., 0.1-1.0 mm

erosion), with a mean of 310 +/- 230 $m^3.km^{-2}$. For these years, variations in landslide rate appear uncorrelated

between catchments, except for 2012 and 2013, which had rather above and below average landslide rates for most

areas, respectively. Notably, we do not find any correlation between measures of monsoon strength derived from

satellite measurements (i.e., GSMaP rainfall estimates, cf., Kubota et al., 2006, Ushio et al., 2009) in each

catchment (total rainfall of days between May or June and October; of days above an intensity threshold during; of

the wettest sequence of 20 or 40 days) and landslide rates (Suppl. Fig. 4). Nevertheless, at the rates observed in the

four mapping areas during the period 2010-2014, 10-20 years of monsoon-induced landsliding would suffice to

match the landsliding caused by the 2015 Gorkha earthquake in the BK, while the 12 May aftershock caused an

amount of landsliding in the BK equivalent to one or two monsoon seasons (Fig. 2). In BG and T, the earthquake-

induced landsliding is equivalent to ~40 to 60 years of the mean landslide rates caused by the 2010-2014 monsoons.

Importantly, the stable average landslide rate, across catchments and through time, was obtained by

excluding the single largest landslides in 2013 and 2015 in KG and in 2014 in BK (Jure landslide). These landslides

are difficult to attribute to any given monsoon season because they appear to have been caused by progressive

destabilization. For the 2013 and 2014 landslides, small scale landsliding occurred around the scarps in preceding

years, while the 2015 landslide was reported to have developed significant cracks at its crest during the earthquake

that year. Further, these landslides depart significantly from the probability density distribution defined by the RE

inventory (Cf . 3.1.4) and we further discuss their origin in 4.1.

Two of the large landslides mentioned above are also identified in our multidecadal mapping from Landsat

images. The 2014 BK (Jure) and 2013 KG landslides feature amongst 49 landslides ranging from 0.08 $km^2$ to about

0.8 $km^2$. After runout correction, their scar areas are between 0.02 $km^2$ and 0.4 $km^2$. They are relatively uniformly

distributed across the whole area of interest (Fig 1). Despite the low resolution of the Landsat imagery, we could

identify in the appropriate time intervals several large failures described in the literature such as reactivation of the Satuiti landslide before the 1990 and between 2002-2011 (cf. Gallo et al., 2014), and the Labubesi (BG, 1968), Dharbang (1988) and Tatopani (KG, 1998) landslides, which each caused notable river damming (Weidinger, 2011). The Satuiti landslide oscillates between slow and rapid downslope movement with widespread collapses during periods of acceleration (Gallo et al., 2014). The river blocking landslides mentioned above are also considered to be at least in part related to specific geomechanical conditions, with important roles for rock mass fabric, stress release and erosion (Weidinger, 2011). The 2014 Jure landslide in BK, is the largest single failure to have occurred in our observation window since 1970, clearly demonstrating that its probability of occurrence would be greatly over-estimated based on its inclusion in the 8 year record from our RE mapping.

### 3.1.3 Earthquake perturbations of monsoon-driven landsliding

The 2015 monsoon season started shortly after the Gorkha earthquake and the large 12 May aftershock and caused exceptional landsliding in the three RE mapping areas (Trisuli, Bhote Koshi and Buri Gandaki) significantly affected by strong ground motion and coseismic landsliding. Landsliding in T, BK and BG reached 400 to 600 m²/km² and 1000 to 2500 m³/km², ~3-6 times the 2010-2014 average (Fig. 2). Only 20-30% of these landslides overlapped with recognized coseismic landslides, implying potential reactivation, confirming that the elevated landslide rate during the 2015 monsoon was due mostly to new landslides in weakened but previously stable slopes, as observed after other earthquakes (Marc et al., 2015). In contrast, at 110 $m^2$ $km^{-2}$ and 180 m³ $km^{-2}$, the landslide rate in KG was slightly below the 2010-2014 average in this area. For other large, shallow earthquakes, elevated propensity to rainfall-induced slope failure has been reported to last from 0.5 to 4 years (Marc et al., 2015). The 2016 monsoon was stronger than usual and solicited above average landsliding in the KG and T but not clearly in the BK and BG (Fig. 2). In 2016, the BK area was also affected by a glacier lake outburst flood that caused intense channel bank erosion and collapse of fringing hillslopes (Cook et al., 2018). Landslide rates in 2016 were two orders of magnitude higher than the pre-earthquake mean in a corridor (i.e., in the lower half of the slopes) along

the Bhote Koshi main stem. However, if all landslides in this corridor are attributed to the flood and not taken into consideration, then the remaining landsliding is below the pre-earthquake average rate of monsoon-driven mass wasting (Fig. 2). In 2017, all catchments were within the pre-earthquake range. Analysing landslide density, that is total number normalized by the mapping area, would yield the same conclusions (Suppl. Fig. 5).

Thus, after the 2015 earthquake, landslide susceptibility was significantly elevated during the 2015 monsoon, but had recovered in 2017. Without an empirical correction for the variability of landsliding due to monsoon strength it is unclear, yet, if the landsliding in 2016 was still affected by the earthquake. For now, we can only delimit the recovery between a few months and 1.5 years. A better understanding of the variability of landsliding in response to monsoon rainfall is required to refine this estimate.


### 3.1.4 Landslide size distributions

To understand the long-term erosion caused by landsliding it is essential to quantify the frequency of small and large landslides, and how it varies through our study area and with rainfall and seismic triggers. Size distributions of monsoon-induced landslide scars exhibit a typical probability density distribution (cf. Stark and

Hovius, 2001), with characteristic power law decay from $10^3$ to $10^5$ m², and a roll-over between 100 and 300 m². Following Malamud et al., (2004), and using a maximum likelihood estimation (MLE), we can fit, an inverse-gamma distribution to each dataset with almost identical mode and scaling exponent (i.e., $P(A) \sim A^{-(\alpha+1)}$ , where P is a probability density function) $\alpha_M + 1 = -2.4$ +/- 0.05 (95% confidence interval from MLE) (Fig. 3). Applying the method of Clauset et al. (2009), we find a power-law tail beyond a threshold area of ~1200m² with $\alpha_M + 1 = 2.48$ +/-

0.1 (1-sigma for 150 bootstrap replicate determinations of $\alpha$). The landslide scar area distribution derived from the catalogue of Roback et al., (2018) can be described by an identical exponent, but with a larger threshold area of ~2500m². We also note that the 2015 landslides in BK, BG and T have similar size distributions to the ones found

for these RE mapping areas in 2010-2014, with $\alpha+1=2.39$ +/-0.12 and $\alpha+1=2.43$ +/-0.18 (Fig. 3). This means that after the earthquake the landslide susceptibility was increased equally at all length scales relevant to mass wasting,
consistent with what has been reported for other earthquakes (Marc et al., 2015).

Finally, we note that for a number of monsoon seasons, the largest landslides seem distinct from the rest of the distribution. This is particularly clear when comparing the scar areas of the largest and 2nd largest landslides for each monsoon season and RE mapping area (Fig. 4). In T and BG, the largest landslide is never more than 3 times larger than the second largest, and for most monsoon seasons their sizes are very similar. In contrast the
largest landslides in the 2013 and 2015 KG and the 2014 and 2016 BK inventories are 10 to 100 times larger than the 2nd largest ones. For the 2016 BK inventory, removing the large bank collapses likely caused by the glacier lake outburst flood resolves this discrepancy. With an adequate sampling of the size-frequency distribution, we would expect the maximum landslide area ($A_{max}$) in a random subset to increase with the total number of landslides in that subset. For an inverse gamma distribution with parameters $\alpha$ and $\beta$, the theoretical total landslide number is $N =$
$\alpha\Gamma(\alpha)(A_{max}/\beta)^{\alpha}$ , with $\Gamma$ is the gamma function (cf Eq. 25 in Malamud et al., 2004). The same expression holds for the second largest landslide, if a prefactor of 2 is added to the right-hand-side of this equation (Malamud et al., 2004). This prediction agrees within a factor of 2 with the size of the 2nd largest landslide scar for almost all monsoon seasons (Fig. 4 inset), but the largest landslide in the subsets with outliers discussed above (i.e., 2013 and 2015 in KG and 2014 and 2016 in BK) would require drawing 10 to 100 times more landslides to be consistent
with this distribution.

### 3.2 Long-term sediment mobilization by landslide

Using essential landslide population characteristics gleaned from our combined data sets, we can now estimate long-term erosion by landsliding due to seismic and monsoon forcing based on the absolute frequency ($yr^{-1}.km^{-2}$) of landslides of all sizes.

## 3.2.1 Frequency of earthquake- and monsoon-induced landslides

Based on the comprehensive inventory of landslide polygons mapped by Roback et al. (2018), the frequency of earthquake-induced landslides varies from $10^{-2}$ $km^{-2}.yr^{-1}$ for the modal scar area of ~300m², to $10^{-6}$ $km^{-2}.yr^{-1}$ for 0.3 km² scars (Fig. 5). The frequency decays with increasing landslide size as a power-law with exponent $\alpha_{EQ}$ ~1.42, in the size range from ~2000 to 300,000m². Note that this is consistent with a probability density function exponent (i.e., $\alpha$+1) of 2.4. Extrapolating this power-law trend to the size of observed giant landslides (10-20 km²), we obtain a frequency of ~2[1-3]. $10^{-9}$ $km^{-2}.yr^{-1}$ (Confidence interval for 1-sigma range of the fitting parameters). This is ~10-30 times lower than our frequency estimate from dated giant landslide deposits (Fig. 5).

To obtain a landslide size-frequency distribution representative of monsoon forcing we exclude the post-seismic period during which landslide susceptibility was elevated. This period appears to have been mostly limited to 2015 and accordingly we use a catalogue describing 7 years of monsoon-induced landslides mapped from RE images. The anomalous mass wasting of the 2015 monsoon could be attributed to earthquake-induced effects. However, including these landslides in the seismic budget is not straightforward and this is kept for discussion. The comprehensive mapping from RE imagery covering the most recent monsoon seasons constrains well the distribution of intermediate size landslides (300m² to 10,000m²), but inadvertently over-estimates the frequency of large (>$10^5$m²) landslides (Fig. 5). The multi-decadal catalogue of large landslides mapped from Landsat images allows extension of the range of the landslide size-frequency distribution. Complementarity of the two monsoon data sets is borne out by the fact that the power-law decay of the RE catalogue, defined between ~1000 and 70,000 $m^2$ by $\alpha_M$~1.5, predicts within ~1-sigma uncertainty the frequency of larger landslides with scar areas of 0.07, 0.1, 0.2 and 0.4 km² (Fig. 5) as determined from the Landsat catalogue. In the latter, smaller landslides exhibit a roll-over likely emerging for mechanical reasons (cf Stark and Guzzetti 2009, Frattini and Crosta 2013, Milledge et al., 2014) given that it occurs for sizes below 500 $m^2$, quite larger than our resolution limit (i.e., a few pixels or ~100 $m^2$ ). Nevertheless, the power law best fit combining both datasets has $\alpha_M$ ~1.55 (consistent with the power law

best fit to the 7-yr RE data and uncertainty obtained following Clauset et al., 2009). Using this scaling exponent, we obtain a frequency of $1.2[0.6\text{-}1.8].10^{-8}$ $km^{-2}.yr^{-1}$ for giant landslides. This is ~3-5 times below the frequency estimates of dated deposits (Fig. 5).

The Holocene giant landslides are not specifically attributed to a trigger mechanism, and their estimated frequency has uncertainties. Nevertheless, expected frequencies of MIL and EQIL alone or summed, do not reach the lower frequency estimates for giant landslides. This implies either that another process is the main driver of giant landsliding, or that we have underestimated the frequency of EQIL and/or MIL, as discussed in section 4.1/4.2.

### 3.2.2 Long-term contributions

Integrating the best-fit frequency from $2000m^2$ to the maximal landslide size, we obtain a long-term erosion rate from EQIL and MIL of 0.1 [0.08-0.14] $mm.yr^{-1}$ and 0.8 [0.6-1.2] $mm.yr^{-1}$, respectively. According to this approach, the total landslide erosion is about 0.9 [0.7-1.3] $mm.yr^{-1}$, with a modest 11% due to EQIL. Given the value of the best fit landslide size-frequency scaling exponent, about 70% of the total landslide erosion in this estimate comes from landslides with scar areas larger than 0.02 km² (~0.3 Mm³), and 40% from ones larger than 0.3 km² (~10 Mm³) (Fig 5A). For both MIL and EQIL, we numerically computed the long-term erosion associated with landslides smaller than 1000 $m^2$ (i.e., in the roll-over of the size-frequency distribution) and found it to be less than 5% of the long-term erosion due to the large landslides following a power-law behaviour. The largest landslide has a frequency of ~$3.10^{-9}$ $km^{-2}.yr^{-1}$ (Fig. 5), implying a mean recurrence time $\tau = 30$ kyr within a 10,000 $km^2$ region. A steady erosion rate is expected for measurements integrating over a few $\tau$, unless the boundary conditions relevant to slope failure change. On shorter time scales, erosion proceeds at spatially and temporally variable rates.

## 4. Discussion

### 4.1 Size-frequency distribution and controls on monsoon-driven landsliding.

The long-term erosion associated with MIL and EQIL was derived with the assumption that the landslide size-frequency distributions defined by the 7-year RE and 46-yr Landsat datasets and by the Gorkha landslide inventory, respectively, are representative for the entire area of interest and for timescales of 10 to 100 kyr. If the landslide size-frequency distribution reflects landscape mechanical and topographic properties (cf. Stark and Guzzetti, 2009, Frattini and Crosta, 2013), then the similarity of the distributions in all datasets supports our earlier assumption that the four RE mapping areas as well as the area affected by coseismic landsliding are not significantly different (in terms of landslide dynamics), and that our wider area of investigation can be considered homogeneous. Within this area, we can quantify the variability due to earthquake activity and estimate the resulting landsliding on 10-100 kyr timescales using existing models, as detailed in the next section. However, on these longer timescales monsoon properties have certainly varied, and it is hard to determine how this may have affected the landslide size-frequency distribution, given that we have not found a connection between monsoon meteorological properties and landslide statistics in the last eight years. This negative result may be due to the rainfall estimate we used, derived from satellite, which do not capture localized intense rainfall event, that may be important for landsliding and explain landslide clusters occurring in different sub-parts of the RE images, from one year to another. Alternatively, annual landsliding may be weakly related to hydro-meteorological properties because of a moderate monsoon variability compared to a system exposed to the extreme weather associated with typhoons, or possibly because preconditioning factors are dominant relative to the rainfall forcing. Indeed, we have observed that recent, large landslides can depart significantly from the size-frequency distribution evaluated over short time scales (Fig. 4), but that they sit well within the regional landslide statistics compiled over longer timescales (Fig. 5). From a mechanistic point of view, the failure of the large 2013 and 2015 KG, and 2014 BK landslides may have been controlled by progressive mechanical weakening (Weidinger, 2011, Lacroix and Amitrano, 2013), rather than by

monsoon-driven pore-pressure changes, which govern the occurrence of shallow landslides in soil and regolith. This would imply that on short timescales the hazard posed by large landslides correlates weakly with the properties of the monsoon, and that on long timescales the power-law tail of the MIL size-frequency distribution may depend

485    more on processes modulating rock mass degradation (e.g., weathering, damage) than on variations of mean or extreme rainfall. These degradation processes operate at long time scales (1-10 kyr, Lacroix and Amitrano, 2003), and if they dominated large-scale landsliding, then they could yield a rather constant size-distribution over the timescales of integration. This may not be true if variations in glacial, tectonic or climatic processes have modulated these degradation processes, spatially or temporally, across the Himalayas. Thus, assessing potential bias in the

490    MIL size distribution and long-term erosion may require quantification of the relative impacts of monsoon properties as well as the progressive degradation of hillslope stability on regional landsliding.

**4. 2 The contribution of earthquake-triggered landslides to long-term erosion**

Accounting for landsliding induced by a Mw 7.9 earthquake, similar to the Gorkha earthquake and with a return time of ~300 years, yields only a modest EQIL contribution (11%) to long term erosion (3.2.2) and an

495    underestimation of the frequency of giant landslides (3.2.1). Even if the uncertainty on the recurrence time of an earthquake of this magnitude is substantial (at least ~50yr), it is not likely to significantly reduce the order of magnitude difference between MIL and EQIL frequency. Neither do the elevated landslide rates that persist for some time after an earthquake. In the case of the Gorkha earthquake, this transient landslide pulse equated to about 4-6 years of monsoon-induced landsliding in a period of about one year (Fig. 2). For a 300 year return time of a

500    Gorkha-sized earthquake (cf., 2.3), this pulse may represent 1.3 to 2% of the long-term MIL, or up to ~13-18% of the long-term EQIL. Although non-negligible, it still leaves EQIL long-term erosion far behind MIL erosion. This may be a fact of nature in the central Nepal Himalayas, but we recognize two potentially significant controls on a larger contribution of EQIL to long-term erosion.

The first control is earthquake size. Both smaller and larger earthquakes than the 2015 Mw7.9 Gorkha earthquake occur along the Himalayan front, triggering substantial landsliding. Examples include the 2011 Mw 6.9 earthquake in Sikkim, the 2005 Mw 7.5 Kashmir earthquake, and the 1950 Mw 8.6 Assam event (e.g., Mathur, 1953, Sato et al., 2007, Chakraborty et al., 2011). To estimate the contribution of earthquakes of all magnitudes compared to the mass wasting due to the 2015 Mw 7.9 event, we combined a Gutenberg-Richter distribution of earthquakes, consistent with seismicity in Nepal (Avouac, 2015), with a seismologically-consistent model for the volume of earthquake-induced landslides (Marc et al., 2016a) and the area within which they occurred (Marc et al., 2017). The model accounts for seismic moment, fault type, source depth and surface topography and predicted the total landslide volume associated with the Gorkha earthquake to within a factor of 2 of the volume estimated from comprehensive landslide maps (Marc et al., 2016a, Martha et al., 2016, Roback et al., 2018). The long-term erosion caused by all earthquakes of a given magnitude along a portion of the Himalayan front can be written as

$$Etot_{mw} = E_{mw}.P(aff)_{mw}.F_{mw},$$

with $E_{mw}$ the mean erosion per earthquake (i.e. total landslide volume divided by affected area), $P(Aff)_{mw}$ the probability that a given unit surface area ($1$ $km^2$) is affected, and $F_{mw}$ the earthquake frequency (Figure 6). Assuming all earthquakes distribute randomly within a portion of the mountain front, i.e., a ~600 km long band spanning from the Siwaliks to the high range (~150 km) with a reference area of $10^5$ $km^2$, we approximate $P(aff)_{mw}$ by the area affected by EQIL over this reference area. We assume that, except for magnitude, all earthquakes are similar to the Gorkha earthquake, occurring on a reverse fault at a depth of 15 km under a landscape with a modal slope of 28°, thus neglecting variations in topography, climate and lithology. The model predicts that rare, large earthquakes (Mw>7.5) do not cause significantly more erosion than frequent intermediate ones (Mw~6.8) because the increase in landslide volume with earthquake size is mainly associated with an increase in affected area not landslide density (Fig. 6). However, each large earthquake represents a considerable fraction of the Himalayan front, while many intermediate size earthquakes are required to cover the same fraction. The final result is that,

intermediate earthquakes (Mw 6.8) dominate the long-term erosion, being ~20, 2 and 4 times more important than earthquakes of Mw 6, Mw 7.9 and Mw 8.6, respectively, but that other earthquake sizes contribute substantially to total long-term erosion. Hence, to obtain the total earthquake contribution we must integrate from Mw ~6 to the maximal earthquake magnitude. The largest Himalayan earthquake on instrumental record is the 1950 Mw 8.6 Assam earthquake, but closure of the tectonic slip budget may well require larger earthquakes of up to Mw 9 or more to occur (Avouac, 2015, Stevens and Avouac, 2016). For maximum earthquake magnitudes of Mw 8.6 and 9, the cumulative contribution of earthquakes to long-term erosion should be about 2.9 and 3.1 times that of Mw 7.9 earthquakes (Fig. 5). In both cases, increasing the Gutenberg-Richter exponent, $b_{GR}$ to 1.1, leads to a larger contribution by small to intermediate earthquakes and an increase of the total EQIL erosion by about 15%. The opposite would be true for smaller values of $b_{GR}$.

The second control on EQIL is earthquake depth. The Gorkha earthquake may also not have been representative, as it was relatively deep (15 km) and did not rupture the surface. In contrast, paleo-seismological investigations have shown that large surface-rupturing earthquakes (>100km long) have occurred along the Himalayan range (Mugnier et al., 2013, Bollinger et al., 2014). Earthquakes shallower than the Gorkha event would likely produce stronger ground motions and thus trigger more landslides, and also potentially more large landslides. This would be consistent with the attribution of giant landslides (>km³) in the Pokhara area to medieval earthquakes (Schwanghart et al., 2016), and suggests that earthquakes may contribute a non-negligible proportion of the largest landslides in the region. Further, analyses of a global database of 11 EQIL inventories showed a linear increase in the exponent of landslide size probability density function, $\alpha_{EQ} +1$, from 1.9 to 3 with seismic source depth from ~3 to 20km (i.e., $d(\alpha_{EQ} +1)/dz \sim 0.065$ ) (Marc et al., 2016a). The landslide population of the Gorkha earthquake has a size-frequency scaling exponent $\alpha_{EQ} +1\sim2.6$ (for whole landslide areas) with a source at 15km, consistent with this trend. The earthquakes in the global database were all larger than Mw 6.5, and accordingly their ground shaking can be considered to be controlled mainly by attenuation. Therefore, a shallower source would yield larger

strong motion, capable of mobilizing deeper and larger landslides (Marc et al., 2016a, Valagussa et al., 2019). Such

difference is especially expected for out-of-sequence earthquakes, propagating on the MCT, while in-sequence

rupture will propagate further South on the MHT flat zone, away from our study area. Nevertheless, depth is only

one of the controls on seismic ground shaking and the resulting proportion of large landslide, and other geophysical

aspect may modulate them, such as stress-drop and rupture dynamics (Causse and Song, 2015).

We propose a quantitative correction of the EQIL size-frequency distribution, accounting for a range of

earthquake magnitudes, post-seismic elevated landsliding, and for a higher proportion of large landslides as a

consequence of stronger ground shaking. The two former effects are modelled as an increased frequency at all sizes

by a factor 3.3, equal to the erosion from all earthquakes Mw 6 to 9 normalized by the erosion caused by Mw 7.9

earthquakes (assuming a source depth of 12.5km and $b_{GR}$=1, Fig 6), and by a factor of 1.15, assuming the proportion

of post-seismic landsliding relative to coseismic landsliding is constant with magnitude. We explore the effect of a

higher proportion of large landslides by computing EQIL long-term erosion with a progressively increasing

proportion of large landslides relative to a fixed frequency of small landslides (Fig. 5). For example, assuming

landslide scar frequency and whole landslide frequency had similar decays for the cases studied by Marc et al.,

(2016a) (as we found to be the case for the Gorkha earthquake), a decrease  from $\alpha_{EQ}$~1.4 to 1.2, could be caused

by source depth reduction from 15 km to 12 km. With these corrections, and for $\alpha_{EQ}$~1.23-1.28, we find that the

EQIL frequency matches the long-term frequency of giant landslides, and that EQIL would contribute 50-58% of

a total erosion of 1.6[1.1-2.4] to 1.9 [1.3-2.8] mm.yr$^{-1}$ (Fig. 5, 7). It being the only range of scenarios matching the

estimated giant landslide frequency, we consider that $\alpha_{EQ}$~1.23-1.28 is most likely to represent long-term

earthquake-induced landsliding. Modelling the landslide erosion associated with repeating earthquake similar to

the Wenchuan earthquake, Li et al., (2017) proposed that EQIL erosion rate amount to 55%-130% of the long-term

fission track exhumation rate. Given exhumation rate also showed a focus to the front of the range, where most

earthquakes and EQIL occur, they considered the long-term erosion to be dominated by EQIL, different from the

rather balanced contribution between seismic and non-seismic forcing that we report (Fig 7). In the Wenchuan area rainfall contributions to landsliding was not constrained and it is unclear if the rainfall there are less effective in

mobilizing landslide than the monsoon, or if their impact was underestimated. Thus, refined estimates of the relative contribution of earthquakes to long-term landslide erosion depend on understanding their ability to trigger very large landslides as well as adequately constraining the contribution of non-seismic landslides.

### 4.3 Implications for erosion rates across different timescales

The stochastic nature of landsliding implies variations of the erosion rate averaged over different timescales, associated with the occasional occurrence of very large slope failures and with variations in the strength of seismic and monsoon forcing. A general caveat is that these rates represent mobilization of bedrock into sediment deposited on lower portions of the hillslope and in channels. In contrast, erosion rates derived from sediment budget and $^{10}$Be refer to the materials transported by the rivers. Small landslides ($As <= 10^4$) have small volumes and

likely deposit relatively fine grained materials (mostly from shallow, weathered soil and regolith) that should be remobilized and transported by rivers within one to a few monsoons. Thus to the extent that ~50% and 90% of our RE catalogue had their largest or second largest landslides size at about $10^4$ m², we likely have short term sediment export on the same order than landslide rates. On millenial timescales, evacuation of sediments must depend on river transport capacity and remobilization of debris on hillslopes, likely linked to hydro-climatic forcing (Pratt-

Sitaula et al., 2004, Cook et al., 2018). Recent modelling study suggest that fast (10-100 yr) evacuation of most of any large landslide deposit should be achievable due to river morphology self-adjustment (Croissant et al., 2017). However, the variable state of export of giant deposits ( >80% preserved for Latamrang and Dhumpu (5 kyr) deposits, but ~25% for the Braga (pre-LGM) deposit, Weidinger, 2006), as well as evidence of substantial sediment storage in the high range (Pratt-Sitaula et al., 2004, Blothe and Korup, 2013, Stolle et al., 2018) suggest complex

evacuation dynamics. As a result, landslide erosion rates may be similar to or significantly larger than $^{10}$Be

depending whether landslide evacuation over the last ~1kyr was efficient or not. Nevertheless, the estimated total modern storage in the central Himalayas is ~100 $km^3$ within an area of $>10^5$ $km^2$ (Blothe and Korup, 2013), equivalent to a mean cover of 1m, or about 500 yr of landslide erosion, while fission track indicate that ~2 mm/yr of erosion have been sustained for 10 Myr or more, clearly indicating that on million year time scales landslide deposit are effectively transported and storage is extremely minor.

We obtain landslide erosion rates that increase across time-scales, from highly stochastic low rates of 0.1-1 $mm.yr^{-1}$ for recent monsoons (Fig. 2) to an expected steady rate of at least 1.2 [0.8-1.7] $mm.yr^{-1}$, but more likely 1.9 [1.1-2.8] $mm.yr^{-1}$ with shallower earthquakes triggering more large landslides than the Gorkha event (Fig. 7), over large areas and on 100 kyr timescales. This range of rates matches independent estimates from fluvial sediment budget on the annual to decadal scale, between 0.2 and 0.6 $mm.yr^{-1}$ (Gabet et al., 2008) on the one hand, and those from fission track, between 1.6-2.6 $mm.yr^{-1}$ (Thiede and Ehlers, 2013), on the other. These rates were determined in the Greater Himalayas in central Nepal. [10]Be-derived erosion estimates in similar zones, mostly ranged between 0.2-2 $mm.yr^{-1}$, (Wobus et al., 2005, Godard et al., 2012, 2014), averaging over ~300-3000 years in catchments typically covering 1/10th of our study area (~1000 m²). These values lie between the short-term and the long-term erosion estimates for landsliding, and they are consistent with an integration of landslide frequency over a landslide size range commensurate with the spatial and temporal scales sampled by the cosmogenic radionuclides. For example, sampling a drainage area of 1000 km² and resolving 500 to 1000 years of erosion is equivalent to integrating up to a landslide frequency of ~1 to $2.10^{-6}$ $km^{-2}.yr^{-1}$, equivalent to a maximum landslide size of ~0.5 to 1 km² (25 to 68 $Mm^3$) for both MIL and EQIL corrected for magnitude distribution (Fig 5). The latter yields an erosion rate dominated by MIL of 0.7[0.5-1] to 0.8[0.6-1.1] mm $yr^{-1}$, for magnitude-corrected EQIL frequency of $\alpha_{EQ}$ = 1.43 and 1.23, respectively. The larger variations around these values found in [10]Be studies may be attributed to variations in the timing and size of the last large landslide in a catchment (in addition to potential bias or mixing issues, e.g, Lupker et al., 2012, Portenga et al., 2015). Although data quantity in different subparts of the orogen is

unequal a similar picture is emerging from other areas (Western, syntax, Sutlej, Sikkim), except perhaps in Bhutan

were long term exhumation from thermochronometry may not be larger than 10 Be (Fig 7, Portenga et al., 2015, Thiede and Ehlers, 2013)

        The general good agreement between our landslide erosion estimates and independent constraints on erosion over time scales ranging from $10^0$ to $10^5$ yr suggests that in the High Himalayas, bedrock landsliding can be considered the principal erosion agent and sediment supply mechanism to river from decadal to geological

timescales. Landslide dominant influence require the hillslopes to be coupled to rivers able to evacuate sediments and maintain steep slopes as it occur in the Himalayas. Our findings are consistent with reports from other active mountain belts that landsliding drives sediment production on decadal to centennial scales (Hovius et al., 1997, Blodgett and Isaacks, 2007, Morin et al., 2018). For the first time, we extend this insight to ~100 kyr timescale.

        Moreover, we show that the stochastic nature of landsliding together with the heavy tail distribution of

landslide scar areas can explain the observed increase in erosion rates from short to long timescales in the Nepal Himalayas and elsewhere (c.f., Kirchner et al., 2001). This is the case as long as the spatial and temporal scales of averaging are short compared to ~$3/f_{max}$, with $f_{max}$ the frequency of the largest possible landslides in a region (Fig. 8). For an area of 10,000km² in the Nepal Himalayas, about 100 kyr are enough for about three of the largest landslides to occur, implying that exhumation rate variations measured by thermochronometry over millions of

years (Thiede and Ehlers, 2013) cannot be due to incomplete sampling of landsliding. Instead, to explain these observations, an actual variation of erosion is required, due for example to changing boundary conditions modulating landslide frequencies and/or other erosion processes. In contrast, typical averaging times of [10]Be methods (~600 years for 1 mm.yr[-1] of erosion) are more than 10 times shorter than the time required for steady long-term landslide erosion in the Himalayas. This is true even for the largest catchments sampled so far, for

example the Ganga river at Harding Bridge, gathering drainage from ~ 200,000 km² of mountain terrain (Lupker et al., 2012). Mountain ranges with very large landslides but with a lower landslide frequency (possibly in the Tian

Shan or the Western Andes) may require even longer timescales for steady landslide erosion. In contrast, reducing the maximum landslide size, for example because of a lower relief or weaker rock mass, or increasing the frequency of giant landslides may reduce the required sampling time by up to a factor of 10 to 100. This may be the case for active mountain ranges such as Taiwan or New Zealand, with steady landsliding averaged over 500-5000 yr for 10,000 km$^2$ source area (Fig. 8A). Still, these settings likely require source areas >10,000 km$^2$, well above the typically sampled catchment size of 1000-5000 km$^2$, for [10]Be methods to properly average erosion, especially because such settings likely have higher erosion rates and thus lower [10]Be sampling times. Exhaustive modelling of the bias of 10Be is beyond the scope of this contribution. Nevertheless, for our case study, the proportion of erosion that can statistically be expected to be missed by [10]Be measurements averaging over 600yr, is ~40-60% for individual mountain catchments, and ~20% for a 10,000 km² source area (Fig 8B). The inadequate averaging time of [10]Be compared to the frequency of large landslide is, therefore, a major caveat in addition to incomplete mixing or sediment storage (Lupker et al., 2012, Dingle et al., 2018). It may explain most of the [10]Be variability across small to intermediate catchments and differences between present and paleo-erosion rates. Last, we note that previous studies that modelled the impact of landslides on [10]Be erosion rates (Niemi et al., 2005, Yanites et al., 2009) concluded that accurate estimates could be achieved for catchments much smaller than indicated by our results (10-10$^2$ km$^2$ vs >10$^4$-10$^5$ km$^2$). Both these previous studies underestimated the required spatio-temporal averaging mainly because they substantially underestimated the largest landslides size, using 1 km$^2$ (0.05 km$^3$) instead of ~40 km$^2$ (10-15 km$^3$). In addition, Niemi et al., (2005) used a heavy-tailed landslide size-frequency distribution with an exponent of $\alpha$=1.1, resulting in a higher frequency of large landslides than that borne out by our data.

In summary, large landslides (>1km$^2$, >70 Mm$^3$) with typical recurrence time of <1 kyr affect <1% of an area of ~10,000km$^2$, but contribute at least 30% and likely up to ~50% (if $\alpha_{EQ} = 1.23$) to long-term (i.e., ~100 kyr) erosion rates. This implies that erosion patterns are extremely heterogeneous on even longer timescales. At shorter

time scales, up to 100 kyr, erosion and sediment sourcing may be much more intense in specific hotspots associated

with large-scale landsliding. We can expect such hotspots to preferentially locate in high-relief areas (Korup et al.,

2007). The occurrence of giant landslides would thus always decrease total relief, providing a geomorphic

mechanism limiting the height of Himalayan peaks. Moreover, the occurrence of large landslides with scar areas

>0.1-1km$^2$, that dominate erosion, is often related to the local evolution of rock mass properties, for example shear

localization, ore mineralization along failure planes, the reactivation of tectonic structures, or progressive

weathering due to focused groundwater circulation (e.g., Weidinger et al., 2002, Lacroix and Amitrano, 2013, Riva

et al., 2018). Thus, although they may occur during the monsoon season or an earthquake (Schwanghart et al.,

2016), giant landslides may rather be controlled by the presence and evolution of geological and topographic

features over longer timescales. Further characterization of the controls on, and drivers of these giant slope failures

should be a priority for future research.

### 5 Conclusion: landslide erosion and processes controlling giant landslides

We have estimated landslide erosion on time scales from years to 100 kyr, based on landslide inventories

capturing the impact of monsoons and the 2015 Mw7.9 Gorkha earthquake. Our estimates match independent

constraints on erosion, on annual, millennial and geological timescales, confirming that bedrock landsliding can be

the principal agent of erosion and sediment supply to rivers in the High Himalayas. Further, we have quantified the

relative contribution of seismic and rainfall triggers, and of frequent and small, and rare and large landslides. We

found that the absolute frequency distributions of landslides triggered by monsoon rainfall and earthquakes are

heavy tailed, causing rare, large landslides to dominate the long-term erosion budget. As a result, earthquakes may

represent from 10% of the long-term erosion budget, if the 2015 Gorkha earthquake is taken as representative of

the long-term earthquake population, up to 50-60% if other earthquakes commonly trigger larger landslides. The

latter is likely, based on a consideration of paleo-seismological evidence and a physically-based model of

earthquake-induced landsliding. It also matches better the observed frequency of giant landslides and the long-term erosion rates from thermochronometric measurements.

We have found that the size distributions of monsoon-induced landslides are identical within error across the central Nepal Himalayas, and also similar to the size distribution of landslides due to the Gorkha earthquake. This supports the idea that landslide size distributions are independent of the specific trigger (Malamud et al., 2004), and set by local topographic and substrate characteristics (Stark and Guzzetti 2009, Frattini and Crosta 2013), which appear to be relatively homogeneous throughout our 10,000 $km^2$ study region. However, potential variations of size distributions with trigger properties (cf. Marc et al., 2016a, 2018, Valagussa et al., 2019) must be further evaluated as they may have a key influence on spatial and temporal variations of long-term landsliding, and on the relative importance of earthquake and rainfall drivers in setting the Himalayan erosion budget.

Finally, the dominant contribution of large and giant landslides to the erosion budget, means that erosion rates estimated on short to intermediate timescales from river load measurements and $^{10}$Be in sediment from small to medium size catchments are insufficient for full understanding of long-term drivers of erosion. Only thermochronometric methods averaging over >100 kyr capture erosion over sufficiently long time scales to be meaningfully compared to long-term controls of erosion such climate and tectonics. In this context, our study highlights the urgent need to identify the primary controls on the location and frequency of giant landslide.

**Acknowledgments**

This study was initiated shortly after the 2015 Gorkha earthquake with GFZ-Potsdam HART (Hazard and Risk Team) support. OM was funded by the French Space Agency (CNES) through the project STREAM-LINE GLIDERS "SaTellite-based Rainfall Measurement and LandslIde detectioN for Global LandslIDE-Rainfall Scaling". RB was funded by the German Federal Ministry of Education and Research through the project SaWaM (Seasonnal Water Management for Semiarid Areas), grant N°02WGR1421. The study includes material, the Rapid Eye satellite imagery, © (2018) PlanetTM. All rights reserved. Data provided on behalf of the German Aerospace Center through funding of the German Federal Ministry of Economy and a

RESA proposal (00165). Landsat images are provided by the USGS through https://earthexplorer.usgs.gov/. The authors gratefully used the global GSMaP rainfall products provided by JAXA (http://sharaku.eorc.jaxa.jp/GSMaP_crest/). The authors thank Emmanuel Gabet and an anonymous reviewer for their constructive reviews.

OM, RB, CA and NH designed the study. RB processed Rapid Eye imagery and ran the automatic classification. OM, RB and 715 LI finalized landslide mapping. OM performed all other analysis. OM wrote the manuscript with input from all authors.

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

| Name (valley) | Tsergo Li (Langtang) | Braga (Marsyangdi) | Dhumpu (Kali Gandaki) | Latamrang (Marsyangdi) | Sabche (Seti) | Dhikur (Marsyangdi) |
|---|---|---|---|---|---|---|
| Volume (km³) | 10-15 | 10-15 | ~3 | ~5 | 4-5 | 1 |
| Age (kyr) | 30-50 | Pre Last Glacial advance | 4 | 5.4 | ~0.5 | Holocene |


Table 1: Summary of the age volume and location of the giant deposits considered in our study area. All of them are considered

single failures, except the Sabche deposits that may have been deposited through 3 main events. See text for more details.

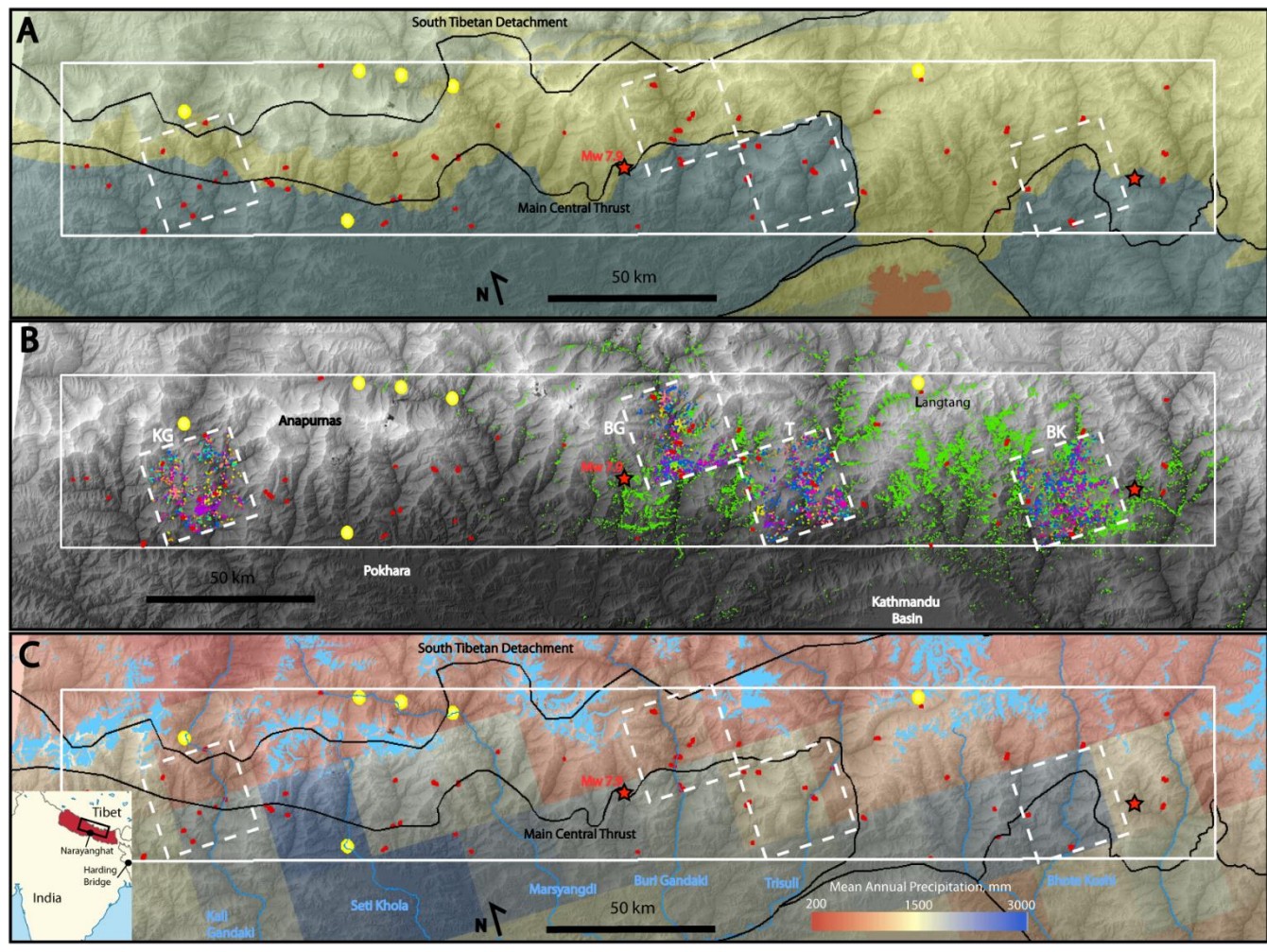


**Figure 1: Hillshaded digital elevation model of central Nepal, with the main geological units (Thetyan sedimentary sequence in grey, High Himalayan sequence in yellow, Lesser Himalayas sequence in blue, Quaternary deposit in red) (A), the different landslide inventories used in this study (B), and the mean annual precipitation, main rivers (blue lines) as well as glacier extents (light blue polygons) (C), within a section of the High Himalayas (white box). In all panel we show the epicenter of the Gorkha earthquake (Mw 7.9) and of its largest aftershock (Mw 7.3) as red stars, the footprint of the Rapid Eye images used to map monsoon-induced landslides from 2010 to 2017 as white dashed boxes. Large (>0.8km²) landslides mapped between 1972 and 2014 are in red and the yellow circles are known giant landslide deposit (>1km³). In (B) we show earthquake-induced landslides reported by Roback et al., (2018), in green and monsoon induced landslides of each year with a separate colour. In A and C the two main fault system are shown with black thick lines. The annual rainfall was estimated from the 0.25° daily rainfall product APHRODITE derived from an extensive gauge network (Yatagai et al., 2012) and the glaciers are from the RGI consortium (2017).**



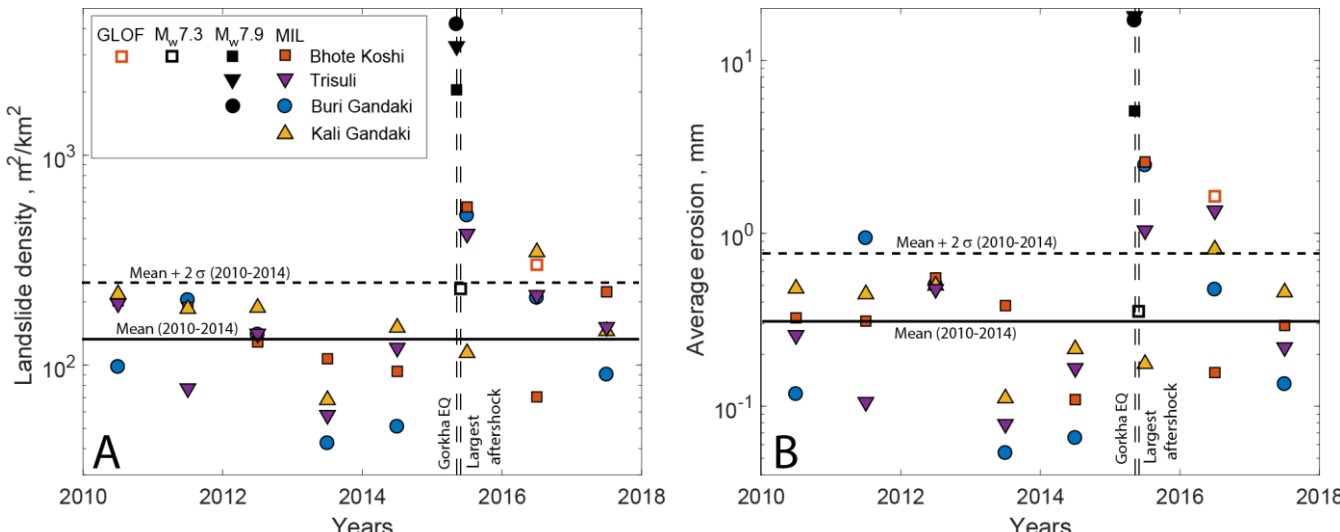

Fig 2 : Landslide density (A) and average erosion (B) associated with the 2010-2017 monsoons in Rapid Eye mapping areas BG, BK, T and KG. Large landslides in KG (2013 and 2015) and BK (2014) have been removed (See text for details). Solid black squares represent the coseismic landsliding due to the Gorkha earthquake in BK, BG and T, while open black square represent the landslides induced by the 12 May 2015 aftershock in the BK valley. Open orange squares indicate the 2016 BK landsliding including bank collapses that are mostly due to aglacier lake outburst flood in that year (Cook et al., 2018). The solid and dashed black lines in A, and B, are the mean values of all catchments and the mean + 2sigma from 2010-2014. Volume conversion leads to 1-sigma uncertainties between 5 and 30% of the total average erosion volume, relatively small compared to the data scatter.

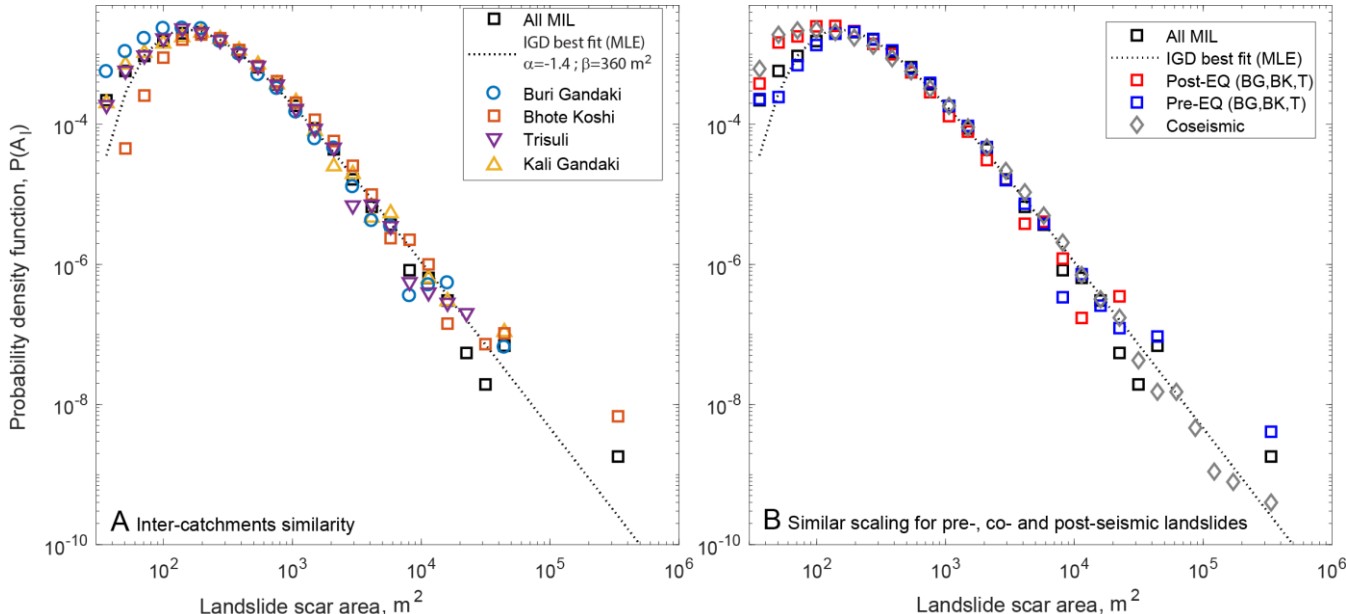

**Fig 3: Probability density functions of landslide scar area for different landslide populations. In both panels, black squares are for the monsoon-induced landslides mapped in the 4 Rapid Eye tiles in the period 2010-2017, and dotted curves show the same best-fit associated Inverse Gamma Distribution. In A, data is subdivided by mapping area. In B, the coseismic landslides (from Roback et al., 2018) normalized for runout, are in grey, while landslides from the monsoon 2010-2014 and 2015, in Buri Gandaki, Bhote Koshi and Trisuli mapping areas are in red and blue respectively.**



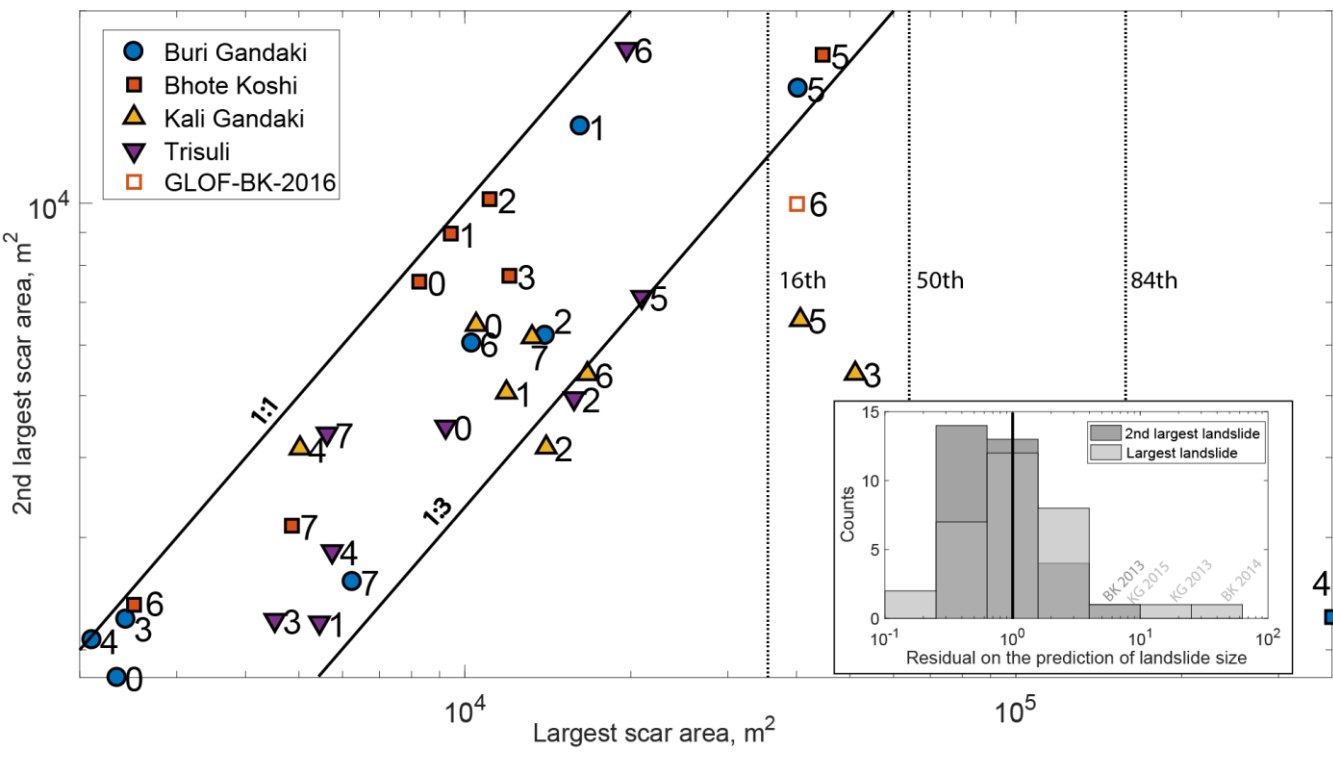


**Fig 4 : Area of the 2nd largest landslide scar plotted against area of the largest landslide scar, for monsoons in the period 2010-2017 and four Rapid Eye mapping areas. The number associated to each symbol indicates the monsoon year relative to 2010. The 2016 Bhote Koshi inventory including the landslides attributed to the glacier lake outburst flood is shown as an open square. 1:1 and 1:3 lines are shown as solid black lines, while the 3 vertical dashed lines indicates the 16th, 50th and 84th percentiles of the landslide scar**
**area from the 46-yr long inventory of landslide with whole area >0.08km². Largest landslides in 2013 KG, 2014 BK and 2015 KG are 10-100 times larger than the rest of the landslide population triggered that year. Inset: Histogram of the residual (ratio) between predicted (as a function of landslide number, cf., Malamud et al., 2004) and observed largest or second largest landslide size. The black vertical line indicate correct prediction. For most years/catchments the predictions are within a factor of 3 of the observed largest size, except for BK 2014, and KG 2013 and 2015. When considering the 2nd largest landslides these sub-inventories become**
**unexceptional.**

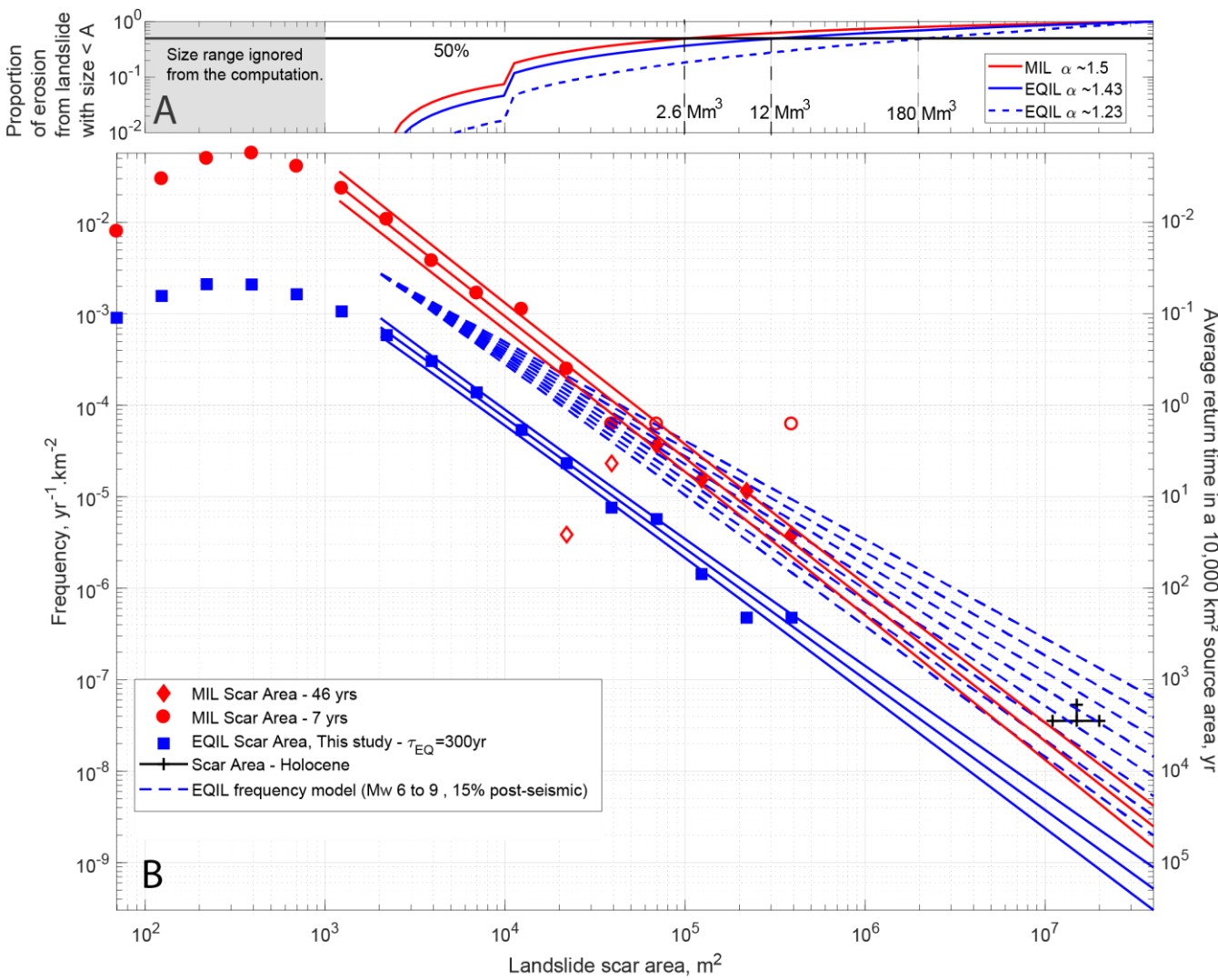


**Fig 5: A:** Proportion of total erosion due to landslide scar larger than a given scar size, against scar size. As a proportion it is independent of the absolute erosion rate (i.e., the landslide mean frequency) but only depends on α, explaining the almost identical curves for MIL (α ~1.5) and EQIL (α ~1.43). **B:** Size-frequency distributions for the scar areas of landslides induced by the 2015 Gorkha earthquake, recent monsoons (2010-2017, except 2015), and large landslides in the last ~46 years. Estimated size and frequency of giant landslides during the Holocene is shown in black. The blue and red lines are the least-square power-law fits with 1-sigma uncertainty range, of the landslide frequency for the Gorkha catalogue and the combined monsoon catalogues (7-years catalogue up to 0.07km² and 46 year catalogue for larger landslides, i.e., ignoring the open symbols), respectively. The blue dashed lines are modelled scenarios for the representative earthquake-induced landslide size-frequency distribution. They include a correction for post-seismic landsliding (+15%) and a factor ~3 increase to account for the contribution of Mw 6 to 9 earthquakes.

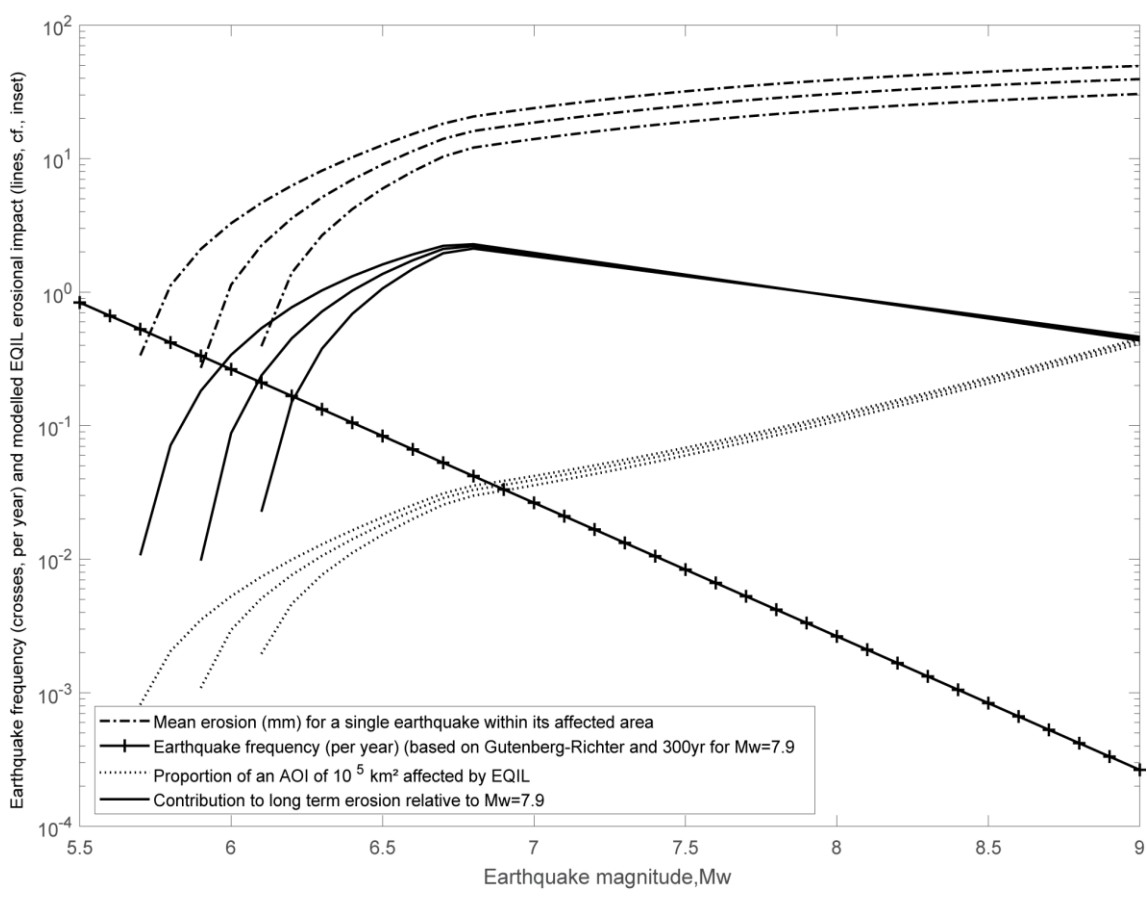

 **Figure 6: Mean landslide erosion (dash-dot line), earthquake contribution to long-term erosion relative to a Mw 7.9 earthquake (solid line), earthquake frequency (crossed line) and earthquake-induced landslide distribution area normalized by a reference area of $10^5$ m² (dotted line), plotted against earthquake magnitude. For each variable the upper, middle and lower curves are for seismic source depth of 10, 12.5 and 15km, respectively.**

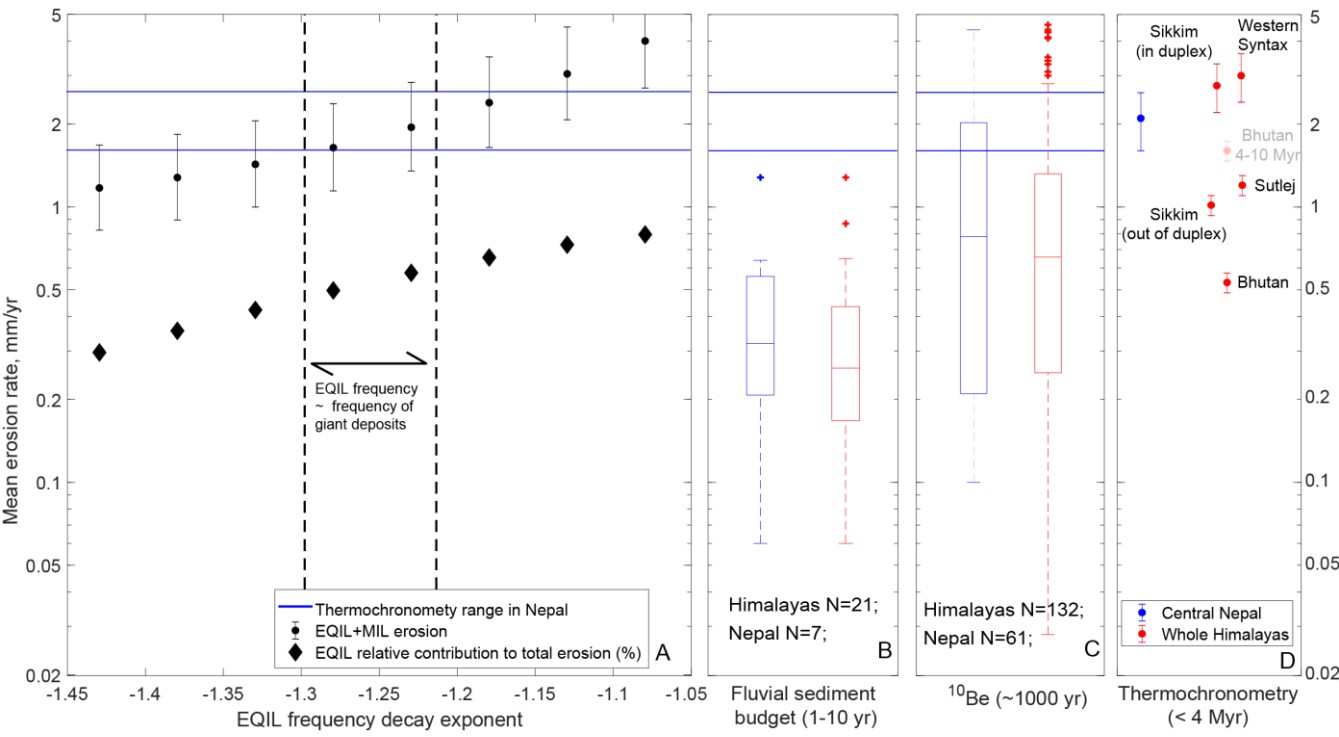

**Fig 7: Long-term erosion rates (circles with uncertainties bar) obtained by integrating and summing the earthquake and monsoon best fit distributions (converted into volume), as a function of the modelled decay exponent of the size distribution of EQIL (A). EQIL distribution takes into account all earthquake magnitudes as well as the post-seismic landslide contribution. The proportion of erosion due to earthquakes in the different scenarios is shown by the black diamonds Erosion rates estimated from fluvial sediment budget (1 to 10 year scale, B), 10 Beryllium catchment wide concentration (1000 year scale, C) and thermochronometric methods (Million year scale, D), in Central Nepal (blue) and the Himalayan arc (red). In A and B, we visualize the data for catchments between 100 and 5000 km², thus excluding main rivers draining large areas. Sediment budgets are from Rao et al. (1997) (Chenab), Ali and De Boer (2007) (Western syntax), Gabet et al., (2008) (Central Nepal), and Wulf et al., (2012) (Sutlej). 10Be measurements are from Wobus et al., (2005) and Godard et al., (2012, 2014) for Central Nepal, Scherler et al., 2014 in the Sutlej, Portenga et al., (2015) in Bhutan, and Abrahami et al., (2016) in Sikkim. Boxplots show 25, 50 and 75 percentiles, whiskers are the furthest data within a distance equal 1.5 times the interquartile range beyond the boxlimit, and data beyond whiskers are shown as crosses. For thermochronometric data we report the mean and standard deviation of the denudation of the models best explaining the age compilation done by Thiede and Ehlers (2013), for the Greater Himalayas sequence in the Western syntax, the Sutlej, Central Nepal and Bhutan. In Sikkim erosion estimates for within and without a zone interpreted as a duplex is from Landry et al., (2016) and Abrahami et al., (2016), respectively.**

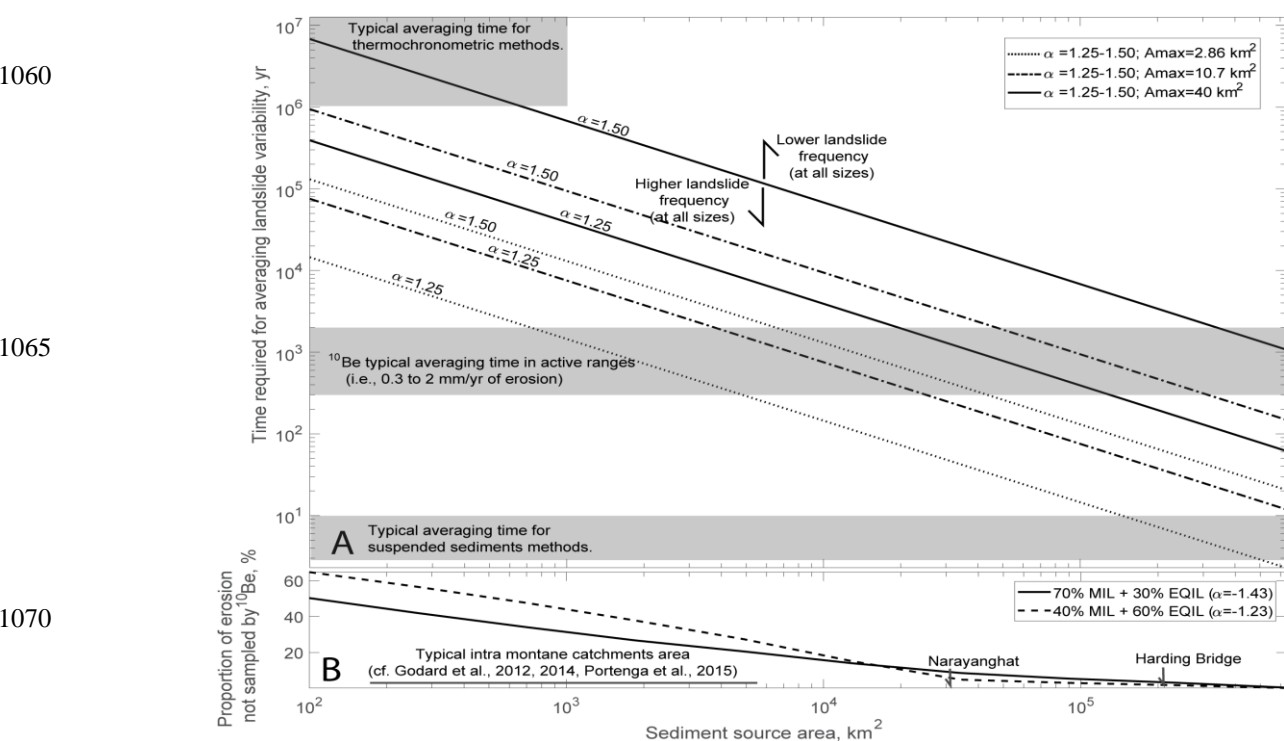

**Fig 8: A:** Estimation of the time required for averaging the statistical variability of landslide erosion (taken as $3/[f_{max}.A_{sed}]$), as a function of the size of the sediment source areas, $A_{sed}$, and the properties of the landslide size-frequency distribution. Typical catchments areas in Himalayan studies, as well as downstream sampling site at Narayanghat or Harding Bridge are indicated, together with the range of averaging time for [10]Be measurements, fluvial sediments and thermochronometric methods. Note that thermochronometric cooling ages are point measurements, but nearby sample are highly correlated up to 10-30km distance (Fox et al., 2016) as long as there are no breaks in tectonic/erosional context (Schildgen et al., 2018). Hence, we consider this methods can be used for spatial scales of ~100-1000 km², consistent with the catchment scales at which detrital thermochronometry seems to be valid (Ruhl and Hodges, 2005). The time scale is inversely proportional with the source areas, but increase strongly with the maximal landslide scar area and the size-frequency power-law exponents ($\alpha$, or equivalently the return time of the largest landslides). Increase or reduction of the overall landslide frequency would result in a proportional changes in the averaging timescale. **B:** Proportion of erosion not sampled by [10]Be measurements averaging over 600 years against the sediment source area sampled. This estimate is based on the proportion of total erosion due to landslide larger than the one with a 600 year return in the Himalayas (Fig 5), considering MIL and Mw-corrected EQIL frequency with a decay similar to the Gorkha earthquake (solid line) or more heavy-tailed (dashed).