# Peer review of "Long-term erosion of the Nepal Himalayas by bedrock landsliding: the role of monsoons, earthquakes and giant landslides."

_Earth Surface Dynamics, 2018_

## Referee Comment (RC1) · E. Gabet (Referee) · 20 Oct 2018

Review of Odin et al. by Manny Gabet

This is an interesting and impressive paper. The authors have compiled a thorough dataset that they use to answer a variety of important questions regarding landslides. Moreover, their analysis seems to have been done with great care. My only quibble is with their assumption that their inventory of small landslides is incomplete, which they use to justify the use of a power-law function to describe the probability distribution of landslide size (as many have done before). As I describe below, I would like to see an analysis of the potential errors associated with this assumption. Otherwise, my

comments are minor.

Comments (keyed to line #)

53 I agree with the general point that the authors are making; however, I'm not sure if it's a good idea to compare Holocene erosion rates with rates averaged over 0-2 Myr to advance the claim that erosion rates increase with measurement time. Clearly, the climate has changed significantly since the Pleistocene and, so attributing the increase in erosion rate to simply a longer measurement period doesn't seem justified unless the climate signal can be accounted for.

77 converted "to" volume

103 It is not clear what the authors mean by "largest single landslides." How does a 'single landslide' differ from just 'a landslide'?

164 the claim that the scars had "fully revegetated" may be a bit strong because it implies that these revegetated areas were indistinguishable from the surrounding areas with respect to plant species, stand age, canopy height, etc; do you mean, instead, that bare ground could no longer be seen?

180 Have you looked at the Tal landslide on the Marsyangdi? If you search for Tal, Nepal in Google Earth, the landslide is about 1 km south of the 'Tal' marker. It is quite large and has completely filled the valley.

189 Please provide a reference for this statement about the terraces.

209 delete "for"?

419 as the authors have done here, the rollover in the pdf at small landslides is typically attributed to incompleteness in the mapping; however, because this rollover is nearly ubiquitous in these types of studies, I think that it would be important to at least entertain the possibility that the rollover is real. The rollover in their data set begins at landslide areas of ∼600 m2 which, given the 5-m resolution of their images, would

be visible (vegetation regrowth notwithstanding). From the standpoint of landslide mechanics, I'm not aware of any physical reason why landslide size ought to obey a strict power-law distribution throughout the entire range of sizes. This means, then, that the authors ought to perform another set of calculations with the assumption that the rollover is real. I understand that this makes the math a little bit more complicated but it is not insurmountable. At the very least, the authors ought to present an estimate of the error associated with assuming a strict power-law distribution if, in fact, the rollover is real.

537 word missing?

556 it is encouraging that your rates are similar to what we found in our 2008 paper but I'm not sure that comparing these is entirely valid, at least without making explicit an important assumption. We were measuring sediment yield and so, to compare your values to ours, you are making the assumption that the total volume of sediment mobilized by all of the landslides was conveyed out of the watersheds. For large landslides, however, we know that this is not true. Also, although we did only measure suspended sediment, we did make a correction for bedload so it might be worth clarifying that our erosion rate was not just from the yield of suspended sediment.

603 this result is sobering and important; I would recommend including it explicitly in the Abstract; my concern is that the effect of landslides is too often ignored in 10Be analyses

Figure 2 I found Figure 2 difficult to interpret because all the markers are darkish. Perhaps reduce the line weight of the markers and choose brighter colors?

---

## Author Comment (AC1) · 23 Oct 2018

We thank Manny Gabet for his positive appraisal of our work.

We will give a detailed answer to all comments when receiving the other review, but here we want to quickly address the major comments made by the referee about the importance of the roll-over.

We agree with the referee that the roll-over may be physical and not due to censoring.

Thus, the sentence P19 L420 : "exhibit a roll-over likely due to incompleteness of mapping associated with the low image resolution and the length of time between successive images" is indeed likely misguiding and we will rephrase it and mention possible physical origin of the roll-over, as described in Stark and Guzzetti, 2009 and Frattini and Crosta 2013.

However, this has no impact of the rest of our analyses, because we have already computed the equivalent erosion due to landslides within the rollover, as stated P19-L437: "The contribution from landslides smaller than 1000m$^2$ (i.e., in the roll-over of the size-frequency distribution) is <5%."

So in our study, given the power law exponent of alpha=1.5 and maximal size of 40km2, the numerical integration of landsldie frequency in the roll-over is <5% of the integral of the power-law behavior between 2000 and 40,000,000 m$^2$. Contribution of the roll-over for the EQIL would even be less with alpha=1.45 or less.

We consider rephrasing slightly this point so it is clear to any reader that we do not assume that the roll-over do not exist but actually compute its contribution and find it negligible. For example: "For both MIL and EQIL, we numerically computed the long-term erosion associated with landslides smaller than 1000m$^2$ (i.e., in the roll-over of the size-frequency distribution) and found it to be less than 5% of the long-term erosion due to the large landslides following a power-law behavior. "

Sincerely, Odin Marc

---

## Referee Comment (RC2) · Anonymous Referee #2 · 26 Oct 2018

This article by Marc and co-authors presents a systematic analysis of landslide size frequency distribution in the Himalaya of central Nepal. They extract these distributions from various datasets, ranging from high resolution imagery over the last few years (covering the period before and after the 2015 Gorkha earthquake) to medium resolution imagery spanning several decades, ultimately incorporating the records of large landslides over the Holocene. One of their focus is the comparison of the distribution parameters and respective landslide volume for different types of forcing such a monsoon precipitation (MIL) and ground shaking associated with the events similar to the 2015 earthquake (EQIL). They also compare the erosion budget associated with landslides over several timescales with integrative estimates of landscapes erosion, derived

from three types of methods : river sediment gauging, detrital CRN and low temperature thermochonology, spanning time frames from a few years to a few Myrs. This is, to my knowledge, the most thorough study of the frequency distribution of landslides over several timescales, in the Himalayas. This study represents a great increment in our understanding of this processes in orogenic systems and its relationship with tectonic and climatic forcings (and in particular their respective influence).

I have one main concern, which I think that the authors can easily address with moderate editing of their discussion. The authors infer landslide erosion rates for their computed distribution that they compare with independent estimates derived from other methods, which allows to point out the relative importance of landsliding (either MIL or EQUIL) in the erosion budget of the range, as well as highlighting important methodological issues with some commonly used approaches (integration time scale of detrital CRN). There is, from my point of view, a major caveat that needs to be explicitly acknowledged and discussed, as it make such comparison very difficult or at least a bit fuzzy. Landslides move large masses of bedrock and regolith on hillslopes, but over relatively small distances at the scale of the range. On the short term (to be defined) the result from a single event is basically a movement of hillslope material at a lower position on the hillslope. There is no doubt that such event will trigger an increase in sediment fluxes due to the production and exposition of easily erodible material, but such spike seems to have a time-span of years to decades from the data available in Taiwan and E Tibet. For large landslides most of the material volume (as calculated from the scaling relationship used here) is going to sit passively on the hillslope for 100s or 1000s of years. Its actual time scale of mobilization is unclear but depends on many factors mostly related to local fluvial dynamics (incision rate at the base of the hillslope, variability in transport capacity etc …), which are very difficult to constrain and makes the actual timescale of the "real" erosion associated with landslides quite open for discussion. This have probably no incidence over thermochonological timescales but should be explicitly addressed when comparing their results with others methods. I think that the study make a very convincing and interesting point concerning

the respective contribution of EQIL and MIL to the global landslide budget over several decades, but I am less convinced by the comparison of the inferred "rates" with other erosion estimates, and I am afraid that putting to much emphasis on this part of the discussion might weaken and blur the message of the paper.

I also have some concerns with the assumption of homogeneity of many climatic and tectonic parameters over the studies area, as it is located across a very strong rock uplift and precipitation gradient. Similarly the preconditioning impact of glacial and periglacial processes in the upper part of the high range should be acknowledged.

Additional comments keyed to line numbers

55-56Âă: this is not clear from the data you present, maybe a supporting figure would help to make your point here

75-76Âă: one or two sentences to explicit the meaning of this cut off size would help understanding the data latter

114Âă: "where the absence of vegetation did not allow mapping" I think I understand why but it should be stated explicitly in this methods part.

126-128 : What about hillslopes with little vegetation cover, especially high in the range? Do you also take into account seasonal effects (e.g. Sal forests)

130 : DEM derivatives, be explicit about what they are (gradient, curvature ?), how they are calculated and what type of information they bring in.

145-146 : not directly the topic of the paper but is there a relationship between these movements and precipitation?

153-155 : I think it's an oversimplification of the context, which is quite heterogeneous from my point of view. The bedrock geology of the northern part of the studied area is probably dominated by the Thetyan series, quite different from the HH gneisses, and they are also intensely sheared (and potentially weak) zones at several positions in the

series, which might affect rock mass properties. One could also note the important variations in rock uplift between the southern and northern parts of the studied area. Also there is a strong climatic gradient across the high range, with most of the precipitation focused on the southern front. Additionally post-glacial debuttressing might be also a process to consider when analyzing the long-term record.

351 : missing )

450-455 : there are evidences of climatically modulated sediment fluxes for many parts of the Himalayan arc (at for the Holocene), so maybe you could discuss the potential influence of changes in landsliding dynamics, as compared to the contribution of glacial erosion in the high range

457 : can they be related to the glacial history in the highest part of the range?

468-470 : what about the influence of short wavelength variations in precipitation (cf Gabet et al., 2004)?

section 4.2 : Globally I am surprised by the absence of comparison of the results obtained here in the Himalayas with the Wenchuan Earthquake and the corresponding discussion of its influence on long-term topographic evolution as, for example, in the following reference : Li, G., West, A. J., Densmore, A. L., Jin, Z., Parker, R. N., & Hilton, R. G. (2014). Seismic mountain building: Landslides associated with the 2008 Wenchuan earthquake in the context of a generalized model for earthquake volume balance. Geochemistry, Geophysics, Geosystems, 15, 833–844. https://doi.org/10.1002/2013GC005067

493 : define "significant portion of the Himalayan front", topography, climatic conditions and strain partitioning are actually quite variable along strike

498 : "mountain front" is not very clear in the context of the Nepal Himalayas, are you referring to the southern part of the High Range?

519 : "Earthquakes shallower than the Gorkha event ..." : but not located at the same

place, a large portion of the seismic moment would be released farther to the south, in the Lesser Himalayas above the MHT flat

4.3 See main concern about the residence time of landslide material in the range, in particular for large events. Some of the arguments presented here seems to rely on the assumption that once landslides occurred, the corresponding material is instantaneously removed and transported away

555-560 : these comparisons between rates from different methods do not make much sense if you do not emphasis the context and particularities of these, as well as give more information about which data are used and their relevance to your area of interest. They seem to encompass very larges and diverse areas both along and across strike. In particular, to what extent are the lower bounds defined by data from the LH, which are actually outside of the investigated area?

558 : Portenga et al. (2005) actually in Bhutan.

559-560 : given the very large reported ranges are these comparison really meaningful?

570-571 : bedrock landsliding will only be an efficient erosion mechanism if river incision is able to maintain local hillslope gradients close to the critical value, and mobilize the corresponding material over the timescale of interest.

576Ăă: "the observed increase in erosion rates from short to long timescales" this is very fuzzy to me, a plot of erosion rates vs integration time scale would probably help (with actual data, not just the ranges). See similar comments above.

Figure 1 : I would expect this introductory situation figure to provide more context concerning the geology and climate of the area. Additional panels (same size and extent) with the corresponding information would be necessary, from my point of view, in particular if you want to support the hypothesis of homogeneity of many of these parameters made above.

[Figure]

Figure 2 : add a vertical bar for the eq(s) date. Lot's of different symbol, some of which are defined in the legend text, not on the figure (orange square). Maybe use a matrix form for the legend (catchments as columns and type of inventories as rows)? Maybe add an upper panel with cumulative monsoon rainfall (same time axis)?

Figure 3 : recall the fitted parameters in the legend. A short title for each panel would help navigation (applicable for other figures)

Figure 4 : for the inset I would draw a vertical bar at 1 (and probably break the bins here)

Figure 5 : I am probably reading that wrong , but for the upper panel Y-axis label should not be "....landslide with size<A"?

---

## Author Comment (AC2) · 8 Dec 2018

We thank the referee for its detailed comments.

We only address here its two main comments.

1/On the evacuation timescales for landsliding:

» The transport dynamics is indeed a difficult issue, that remain an important challenge for the community. We now added a paragraph to detail explicitly transport issues:

" A general caveat is that these rates represent mobilization of bedrock into sediment deposited on lower portions of the hillslope and in channels. In contrast, erosion

rates derived from sediment budget and 10Be refer to the materials transported by the rivers. Small landslides (As<=10^4) have small volumes and likely deposit relatively fine grained materials (mostly from shallow, weathered soil and regolith) that should be remobilized and transported by rivers within one to a few monsoons. Thus to the extent that ~50% and 90% of our RE catalogue had their largest or second largest landslides size at about 10^4 m$^2$, we likely have short term sediment export on the same order than landslide rates. On millenial timescales, evacuation of sediments must depends on river transport capacity and remobilization of debris on hillslopes, likely linked to hydro-climatic forcings (Pratt-Sitaula et al., 2004, Cook et al., 2018). Recent modelling study suggest that fast (10-100 yr) evacuation of most of any large landslide deposit should be achievable due to river morphology self-adjustment (Croissant et al., 2017). However, the variable state of export of giant deposits ( >80% preserved for Latamrang and Dhumpu (5 kyr) deposits, but ~25% for the Braga (pre LGM) deposit, Weidinger, 2006), as well as evidence of substantial sediment storage in the high range (Pratt-Sitaula et al., 2004, Blothe and Korup, 2013, Stolle et al., 2018) suggest complex evacuation dynamics. As a result, landslide erosion rates may be similar to or significantly larger than 10Be depending whether landslide evacuation over the last ~1kyr was efficient or not. Nevertheless, the estimated total modern storage in the central Himalayas is ~100 km3 within an area of >105 km2 (Blothe and Korup, 2013), equivalent to a mean cover of 1m, or about 500 yr of landslide erosion, while fission track indicate that >1 mm/yr of erosion have been sustained for 10 Myr or more, clearly indicating that on million year time scales landslide deposit are effectively transported and storage is extremely minor."

2/ On the homogenity of various environmental factors.

» We agree with the referee that the area is not necessarily homogeneous, however, our results in terms of size distribution or amount of landslides are very similar, suggesting, that as far as landslides are concerned the different zones are similar. Nevertheless we rephrase the methods to : "This area encompasses several lithological

units, a climatic gradient (with enhanced precipitation south of the high peaks, and a rain shadow behind), localized glaciated areas and a likely uplift gradient (Fig S1 or 1). However, the overall result of these heterogeneities on landsliding is unclear and we start by assuming subparts of our study area (e.g., RE tiles, region of coseismic landsliding) have asimilar behaviour and can be compared applying only an areal normalization, and will discuss the validity and caveats of this assumption at the end."

We also added a geological map and climatic map to Figure 1. It is important to note we do not state that these parameters are not varying, only that their net effects on landslide dynamics is not varying much. This does not say that these geographical variations may not matter for other processes, such as soil formation, underground water storage, river flow. . .

All other details comment are answered in our final reply.

---

## Author Response (AR1)

Review of Odin et al. by Manny Gabet
This is an interesting and impressive paper. The authors have compiled a thorough dataset that they use to answer a variety of important questions regarding landslides. Moreover, their analysis seems to have been done with great care. My only quibble is with their assumption that their inventory of small landslides is incomplete, which they use to justify the use of a power-law function to describe the probability distribution of landslide size (as many have done before). As I describe below, I would like to see an analysis of the potential errors associated with this assumption. Otherwise, my comments are minor.
>> This is answered below with the relevant inline comment.

Comments (keyed to line #)
I agree with the general point that the authors are making; however, I'm not sure if it's a good idea to compare Holocene erosion rates with rates averaged over 0-2 Myr to advance the claim that erosion rates increase with measurement time. Clearly, the climate has changed significantly since the Pleistocene and, so attributing the increase in erosion rate to simply a longer measurement period doesn't seem justified unless the climate signal can be accounted for.
>> We understand the reviewer concern. However, we feel presenting both option is nuanced enough because in after correcting for measurement period bias, the need to invoke climatic may disappear. We also note that, even if scatter is large a shift towards large erosion rates seems to be present between the sediment budget and 10Be while no great climatic shift occurred in the last 1000 yr.
Thus we rephrased to: "Although well established, the origin of these features is poorly understood and may be attributed to an inadequate average of extreme events over short timescales, even if climatic variations since the Pleistocene may also have modulated erosion."

converted "to" volume
>> Ok

It is not clear what the authors mean by "largest single landslides." How does a 'single landslide' differ from just 'a landslide'?
>> We removed "single".

the claim that the scars had "fully revegetated" may be a bit strong because it implies that these revegetated areas were indistinguishable from the surrounding areas with respect to plant species, stand age, canopy height, etc; do you mean, instead, that bare ground could no longer be seen?
>> Correct. We mean new vegetation made it similar to the background, but this does not imply a full vegetation recovery.
We now state "revegetation rates (i.e., the time required for vegetation to colonize most of scarp and make it indistinguishable from the surroundings in the available imagery)". Of course this time is not representing the vegetation full recovery.

Have you looked at the Tal landslide on the Marsyangdi? If you search for Tal, Nepal in Google Earth, the landslide is about 1 km south of the 'Tal' marker. It is quite large and has completely filled the valley.
>> The Tal landslide is listed in Weidinger, 2011 as a recent (<1000 yr) Landslide of ~ 5 Mm3 so 200 times smaller than the giant deposits we list (>1km3). It is clear that a number of large (say from 0.01 to 1 km3) have been detected and studied, but it is very difficult to hope to have a comprehensive catalogue over a long period, so we prefer to limit ourself strictly to the largest deposits.

Please provide a reference for this statement about the terraces.

>>We rephrase to: "Other massive terrace deposits in valleys in the High Himalayas result from catastrophic sedimentary events (e.g., Cenderelli and Wohl, 1998, Pratt-Sitaula et al., 2007, Lave et al., 2017), but their conditions of formations are diverse (Glacial lake outburst floods, multiple debris flow, giant landslide evacuation) and relating them to individual landslides challenging."

delete "for"?

>>No, the for is important as slides without long runout are would not be expected to be biased compared to the average A-V relationships.

as the authors have done here, the rollover in the pdf at small landslides is typically attributed to incompleteness in the mapping; however, because this rollover is nearly ubiquitous in these types of studies, I think that it would be important to at least entertain the possibility that the rollover is real. The rollover in their data set begins at landslide areas of 600 m2 which, given the 5-m resolution of their images, wouldbe visible (vegetation regrowth notwithstanding). From the standpoint of landslide mechanics, I'm not aware of any physical reason why landslide size ought to obey a strict power-law distribution throughout the entire range of sizes. This means, then, that the authors ought to perform another set of calculations with the assumption that the rollover is real. I understand that this makes the math a little bit more complicated but it is not insurmountable. At the very least, the authors ought to present an estimate of the error associated with assuming a strict power-law distribution if, in fact, the rollover is real.

>>As we aswered in the open discussion we agree that roll-over may be real and have already computed its contribution to be negligible compared to the landslide sizes with a power-law behaviour.
To make this clearer we rephrased:
The roll-over statement L419 : "a roll-over likely emerging for mechanical reasons (cf Stark and Guzzetti 2009, Frattini and Crosta 2013, Milledge et al., 2014) given that it occurs for sizes below 500 m, quite larger than our resolution limit (i.e., a few pixels or ~100m$^2$ )."

The statement about the contribution of the roll-over L437:"For both MIL and EQIL, we numerically computed the longterm erosion associated with landslides smaller than 1000m2 (i.e., in the roll-over of the size-frequency distribution) and found it to be less than 5% of the long-term erosion due to the large landslides following a power-law behaviour. "

word missing?

>> The original statement had some missing terms, added here in bold and italics: "We explore the effect *of* a higher proportion of large landslides by computing EQIL long-term erosion with a progressively increasing proportion *of large landslides* relative to a fixed frequency of small landslides (Fig. 5)."

it is encouraging that your rates are similar to what we found in our 2008 paper but I'm not sure that comparing these is entirely valid, at least without making explicit an important assumption. We were measuring sediment yield and so, to compare your values to ours, you are making the assumption that the total volume of sediment mobilized by all of the landslides was conveyed out of the watersheds. For large landslides, however, we know that this is not true. Also, although we did only measure suspended sediment, we did make a correction for bedload so it might be worth clarifying that our erosion rate was not just from the yield of suspended sediment.

>> First, we rephrased to "fluvial sediment budget" to avoid suggesting it is only based on suspended sediments. Then, consistently with Referee 2 suggestions we will add a paragraph about landslide deposit evacuation, and discuss how it makes comparison somewhat uncertain.
" A general caveat is that these rates represent mobilization of bedrock into sediment deposited on lower portions of the hillslope and in channels. In contrast, erosion rates derived from sediment budget and $^{10}$Be refer to the materials transported by the rivers. Small landslides ($As<=10^4$) have small volumes and likely deposit relatively fine grained materials (mostly from shallow, weathered soil and regolith) that should be remobilized and transported by rivers within one to a few monsoons. Thus to the extent that ~50% and 90% of our RE catalogue had their largest or second largest landslides size at about $10^4$ m², we likely have short term sediment export on the same order than landslide rates. On millenial timescales, evacuation of sediments must depends on river transport capacity and remobilization of debris on hillslopes, likely linked to hydro-climatic forcings (Pratt-Sitaula et al., 2004, Cook et al., 2018). Recent modelling study suggest that fast (10-100 yr) evacuation of most of any large landslide deposit should be achievable due to river morphology self-adjustment (Croissant et al., 2017). However, the variable state of export of giant deposits ( >80% preserved for Latamrang and Dhumpu (5 kyr) deposits, but ~25% for the Braga (pre LGM) deposit, Weidinger, 2006), as well as evidence of substantial sediment storage in the high range (Pratt-Sitaula et al., 2004, Blothe and Korup, 2013, Stolle et al., 2018) suggest complex evacuation dynamics.  As a result, landslide erosion rates may be similar to or significantly larger than $^{10}$Be whether landslide evacuation during ~1kyr was efficient or not, respectively.  Nevertheless, the estimated total modern storage in the central himalaya is ~100 km$^3$ within an area of >$10^5$ km$^2$ (Blothe and Korup, 2013), equivalent to a mean cover of 1m, or about 500 yr of landslide erosion, while fission track indicate that ~2 mm/yr of erosion have been sustained for 10 Myr or more, clearly indicating that on million year time scales landslide deposit are effectively transported and storage is extremely minor."

this result is sobering and important; I would recommend including it explicitly in the Abstract; my concern is that the effect of landslides is too often ignored in 10Be analyses
>> Following the referee advice we propose to rewrite the last sentence in the abstract:
 This observation presents a strong caveat when interpreting spatial or temporal variability of erosion rates from this method. Thus, in areas where very large, rare landslide contributes heavily to long-term erosion (as the Himalayas), we recommend 10 Be sample on catchment with source areas >= 10,000 km² to reduce the method mean bias below 20% of the long-term erosion.

Figure 2 I found Figure 2 difficult to interpret because all the markers are darkish.
Perhaps reduce the line weight of the markers and choose brighter colors?
>> Ok we reduced symbol lines thickness.

**Anonymous Referee #2**

This article by Marc and co-authors presents a systematic analysis of landslide size frequency distribution in the Himalaya of central Nepal. They extract these distributions from various datasets, ranging from high resolution imagery over the last few years (covering the period before and after the 2015 Gorkha earthquake) to medium resolution imagery spanning several decades, ultimately incorporating the records of large landslides over the Holocene. One of their focus is the comparison of the distribution parameters and respective landslide volume for different types of forcing such a monsoon precipitation (MIL) and ground shaking associated with the events similar to the 2015 earthquake (EQIL). They also compare the erosion budget associated with landslides over several timescales with integrative estimates of landscapes erosion, derivedfrom three types of methods : river sediment gauging, detrital CRN and low temperature thermochonology, spanning time frames from a few years to a few Myrs. This is, to my knowledge, the most thorough study of the frequency distribution of landslides over several timescales, in the Himalayas. This study represents a great increment in our understanding of this processes in orogenic systems and its relationship with tectonic and climatic forcings (and in particular their respective influence).

I have one main concern, which I think that the authors can easily address with moderate editing of their discussion. The authors infer landslide erosion rates for their computed distribution that they compare with independent estimates derived from other methods, which allows to point out the relative importance of landsliding (either MIL or EQUIL) in the erosion budget of the range, as well as highlighting important methodological issues with some commonly used approaches (integration time scale of detrital CRN). There is, from my point of view, a major caveat that needs to be explicitly acknowledged and discussed, as it make such comparison very difficult or at least a bit fuzzy. Landslides move large masses of bedrock and regolith on hillslopes, but over relatively small distances at the scale of the range. On the short term (to be defined) the result from a single event is basically a movement of hillslope material at a lower position on the hillslope. There is no doubt that such event will trigger an increase in sediment fluxes due to the production and exposition of easily erodible material, but such spike seems to have a time-span of years to decades from the data available in Taiwan and E Tibet. For large landslides most of the material volume (as calculated from the scaling relationship used here) is going to sit passively on the hillslope for 100s or 1000s of years. Its actual time scale of mobilization is unclear but depends on many factors mostly related to local fluvial dynamics (incision rate at the base of the hillslope, variability in transport capacity etc : : :), which are very difficult to constrain and makes the actual timescale of the "real" erosion associated with landslides quite open for discussion. This have probably no incidence over thermochonological timescales but should be explicitly addressed when comparing their results with others methods. I think that the study make a very convincing and interesting point concerning the respective contribution of EQIL and MIL to the global landslide budget over several decades, but I am less convinced by the comparison of the inferred "rates" with other erosion estimates, and I am afraid that putting to much emphasis on this part of the discussion might weaken and blur the message of the paper. I also have some concerns with the assumption of homogeneity of many climatic and tectonic parameters over the studies area, as it is located across a very strong rock uplift and precipitation gradient. Similarly the preconditioning impact of glacial and periglacial processes in the upper part of the high range should be acknowledged.
Additional comments keyed to line numbers

55-56: this is not clear from the data you present, maybe a supporting figure would help to make your point here
>> We have made a new figure compiling fluvial sediment budget, 10Be denudation and fission tracks in Central Nepal as well as in other parts of the range. This figure extends figure 7 where only data ranges where shown before. We also updated the caption as follow:

Figure 7 : Long-term erosion rates (circles with uncertainties bar) obtained by integrating and summing the earthquake and monsoon best fit distributions (converted into volume), as a function of the modelled decay exponent of the size distribution of EQIL (A). EQIL distribution takes into account all earthquake magnitudes as well as the post-seismic landslide contribution. The proportion of erosion due to earthquakes in the different scenarios is shown by the black diamonds.
Erosion rates estimated from fluvial sediment budget (1 to 10 year scale, B), 10 Beryllium catchment wide concentration (1000 year scale, C) and thermochronometric methods (Million year scale, D), in Central Nepal (blue) and the Himalayan arc (red). In A and B, we visualize the data are for catchments between 100 and 5000 km², thus excluding main rivers draining large areas. Sediment budgets are from Rao et al. (1997) (Chenab), Ali and De Boer (2007) (Western syntax), Gabet et al., (2008) (Central Nepal), and Wulf et al., (2012) (Sutlej). [10]Be measurements are from Wobus et al., (2005) and Godard et al., (2012, 2014) for Central Nepal, Scherler et al., 2014 in the Sutlej, Portenga et al., (2015) in Bhutan, and Abrahami et al., (2016) in Sikkim. Boxplots show 25, 50 and 75 percentiles, whiskers are the furthest data within a distance equal 1.5 times the interquartile range beyond the boxlimit, and data beyond whiskers are shown as crosses. For thermochronometric data we report the mean and standard deviation of the denudation of the models best explaining the age compilation done by Thiede and

Ehlers (2013), for the Greater Himalayas sequence in the Western syntax, the Sutlej, Central Nepal and Bhutan. In Sikkim erosion estimates for within and without a zone interpreted as a duplex is from Landry et al., (2016) and Abrahami et al., (2016), respectively.

[Figure]

75-76: one or two sentences to explicit the meaning of this cut off size would help understanding the data latter
>> We added the following sentence: The roll-over and divergence from power law behaviour has been interpreted as due to resolution censoring (Stark and Hovius 2001) or as emerging for mechanical reasons (cf Stark and Guzzetti 2009, Frattini and Crosta 2013, Milledge et al., 2014).

114: "where the absence of vegetation did not allow mapping" I think I understand why but it should be stated explicitly in this methods part.
>> We rephrased to : "where the absence of vegetation did not allow mapping. Indeed, the change from a vegetation signature to a rock debris signature is very conspicuous in multispectral imagery, even for sparse vegetation, whereas textural or spectral changes in rocky/sedimentary surfaces remain challenging to detect and interpret. "

126-128 : What about hillslopes with little vegetation cover, especially high in the range? Do you also take into account seasonal effects (e.g. Sal forests)
>> The algorithm is able to detect disturbances also in less dense vegetation cover. As long as there is a vegetation signal present in the spectrum of the pixel (which is also the case for sparse vegetation) there is the potential for detecting these landslides. The approach classifies the vegetation cover disturbances in multiple classes of disturbance severity. The more severe the disturbance the higher is the probability that this pixel is a landslide pixel. The landslide polygons which are generated based on the vegetation disturbance pixels include pixel of higher and less disturbed vegetation.
If the vegetation cover is not existent any more in the very high ranges the algorithm detects no landslides. However, these barren high altitude zones (often snow-covered in near winter images) are excluded from the analysis as peri-glacial areas, as stated in the text (Cf previous comment).
We do not take seasonal effects into account because we have not enough remote sensing data to build a model which can account for seasonal patterns (e.g. annual vegetation development, phenologic parameters). However, for the processing we have used, when possible, remote sensing images of the same season, in the post-monsoon the season of highest vegetation cover, which minimizes seasonal effects.

We added in the methods : "Limited amount of imagery did not allow for accounting for and removing seasonal variations in the NDVI signatures, but most of the scenes are in the post-monsoon season when vegetation cover is highest, limiting such variations (Table S1)."

: DEM derivatives, be explicit about what they are (gradient, curvature ?), how they are calculated and what type of information they bring in.
>> The DEM derivatives are slope gradient and parallelism to stream flows and are calculated for each landslide candidate object, which are derived by the previous bi-temporal vegetation disturbance and the multi-temporal revegetation analysis. The slope parameter excludes False Positive (FPs) objects that are located on gentle terrain, such as new streets or buildings and remaining harvested fields. The parallelism parameter excludes FPs that are caused by flooded rivers or local co-registration errors (mostly occurring in steep valleys).

We rephrased:  "In combination with slope gradient and parallelism to rivers, which enhance the exclusion of anthropogenic (building, field clearings) and flood related disturbance, respectively, this approach enables automated identification of landslides of different sizes and shapes"

145-146 : not directly the topic of the paper but is there a relationship between these movements and precipitation?
>> This is unclear given our rainfall estimates and the fact that we  have not tried to quantify properly the slow motion of these hillslopes. At least 2 of these blocks show significant movement during the 2009 to 2013 or 2014 period but less clearly in the last year of the record, including in 2015. We added the following sentence: "Potential link between annual movements and the monsoon rainfall are unclear and further investigation would require proper quantification of the block movement history, out of the scope of this work.  "

153-155 : I think it's an oversimplification of the context, which is quite heterogeneous from my point of view. The bedrock geology of the northern part of the studied area is probably dominated by the Thetyan series, quite different from the HH gneisses, and they are also intensely sheared (and potentially weak) zones at several positions in the series, which might affect rock mass properties. One could also note the important variations in rock uplift between the southern and northern parts of the studied area. Also there is a strong climatic gradient across the high range, with most of the precipitation focused on the southern front. Additionally post-glacial debuttressing might be also a process to consider when analyzing the long-term record.
>> We agree with the referee that the area is not necessarily homogeneous, however, our results in terms of size distribution or amount of landslides are very similar, suggesting, that as far as landslides are concerned the different zones are similar.
Nevertheless we rephrase to : "This area encompasses several lithological units, a climatic gradient (with enhanced precipitation south of the high peaks, and a rain shadow behind), localized glaciated areas and a likely uplift gradient (Fig S1 or 1). However, the overall result of these heterogeneities on landsliding is unclear and we start by assuming subparts of our study area (e.g., RE tiles, region of coseismic landsliding) have asimilar behaviour and can be compared applying only an areal normalization, and will discuss the validity and caveats of this assumption at the end."

: missing )
>> Ok.

450-455 : there are evidences of climatically modulated sediment fluxes for many parts of the Himalayan arc (at for the Holocene), so maybe you could discuss the potential influence of changes in landsliding dynamics, as compared to the contribution of glacial erosion in the high range

>> We already explicitly acknowledge variations of the monsoon in the past and their potential impacts in this paragraph.

In contrast, we do not think glacial erosion matters for the orogeny scale budget, and have no data to discuss their local influence on landsliding. To make this focus clear, we added in the introduction: ". In the Himalayas, glaciers do not seem to contribute much to the erosion budget of the range (Morin et al., 2015), likely because in spite of having significant local effects on the erosion dynamics (e.g., Heimsath and McGlyyn, 2008) they have a very limited areal extent, even during ice ages. Thus, we consider that quantitative understanding of role and behavior of landsliding in the Himalayas can be obtained without investigating glacial and peri-glacial areas."

: can they be related to the glacial history in the highest part of the range?
>> We do not know and are not aware of studies in the Himalayas that have looked into the details of the relations between glacial history and rock degradation and or large scale landsliding.
 We have added: "This may not be true if variations in glacial, tectonic or climatic processes have modulated these degradation processes, spatially or temporally, across the Himalayas."

468-470 : what about the influence of short wavelength variations in precipitation (cf Gabet et al., 2004)?
>>It is indeed a relevant point. Localized intense rainfall episods during the monsoon may explain landslides spatial clusters occurring in different subparts of our images from one year to another. However, we currently cannot constrain the occurrence of such localized rainfall events and whether or not they agree with our landslide observations.
We added L 475 : "This negative result may be due to the rainfall estimate we used, derived from satellite, which do not capture localized intense rainfall event, that may be important for landsliding and explain landslide clusters occurring in different sub-parts of the RE images, from one year to another."

section 4.2 : Globally I am surprised by the absence of comparison of the results obtained here in the Himalayas with the Wenchuan Earthquake and the corresponding discussion of its influence on long-term topographic evolution as, for example, in the following reference : Li, G., West, A. J., Densmore, A. L., Jin, Z., Parker, R. N., & Hilton, R. G. (2014). Seismic mountain building: Landslides associated with the 2008 Wenchuan earthquake in the context of a generalized model for earthquake volume balance. Geochemistry, Geophysics, Geosystems, 15, 833–844.
https://doi.org/10.1002/2013GC005067
>> We think the topic about the role of earthquake in topographic building in the Himalayas is off the scope of this work, indeed it would require much more discussion about earthquake-induced uplift and interseismic uplift. However, Li et al. (2017) also modelled the long-term impact of EQIL and compared it to erosion in the Wenchuan area. They proposed that EQIL is a significant driver of erosion at the mountain front of the Wenchuan area, based on EQIL inventories. To compare to their results we added
L565: "Modelling the landslide erosion associated with repeating earthquake similar to the Wenchuan earthquake, Li et al. (2017) proposed that EQIL erosion rate amount to 55%-130% of the long-term fission track exhumation rate. Given exhumation rate also showed a focus to the front of the range, where most earthquakes and EQIL occur, they considered the long-term erosion to be dominated by EQIL, different from the rather balanced contribution between seismic and non-seismic forcing that we report (Fig 7). In the Wenchuan area rainfall contributions to landsliding was not constrained and it is unclear if the rainfall there are less effective in mobilizing landslide than the monsoon, or if their impact was underestimated. Thus, refined estimates of the relative contribution of earthquakes to long-term landslide erosion depend on understanding their ability to trigger very large landslides as well as adequately constraining the contribution of non-seismic landslides."

: define "significant portion of the Himalayan front", topography, climatic conditions
and strain partitioning are actually quite variable along strike
>> We simply mean a substantial band above the MHT flat and ramp, "i.e. a ~600 km long band spanning from the Siwaliks to the high range (~150 km) with a reference area of $10^5$ km²".
This stay within the bound of Nepal, only extending our study area of ~ 150 km on both along strike directions and towards the lesser Himalayas and the foothills.
We specify further that assuming all earthquakes are similar means that we are " neglecting variations in topography, climate and lithology "

: "mountain front" is not very clear in the context of the Nepal Himalayas, are you
referring to the southern part of the High Range?
>> Not specifically. We refer to the reference area, mentioned above, and clarified in the reply to the previous comment.

: "Earthquakes shallower than the Gorkha event ..." : but not located at the same place, a large portion of the seismic moment would be released farther to the south, in the Lesser Himalayas above the MHT flat.
>> This true and we now acknowledge it L550 : "Such difference is especially expected for out-of-sequence earthquakes, propagating on the MCT, while in-sequence rupture will propagate further South on the MHT flat zone, away from our study area. Nevertheless, depth is only one of the controls on seismic ground shaking and the resulting proportion of large landslide, and other geophysical aspect may modulate them, such as stress-drop and rupture dynamics (Causse and Song, 2015)."

4.3 See main concern about the residence time of landslide material in the range, in
particular for large events. Some of the arguments presented here seems to rely on
the assumption that once landslides occurred, the corresponding material is instantaneously
removed and transported away
>> The transport dynamics is indeed a difficult issue, that remain an important challenge for the community.
We now added a paragraph to detail explicitly transport issues:

" A general caveat is that these rates represent mobilization of bedrock into sediment deposited on lower portions of the hillslope and in channels. In contrast, erosion rates derived from sediment budget and $^{10}$Be refer to the materials transported by the rivers. Small landslides (As<=$10^4$) have small volumes and likely deposit relatively fine grained materials (mostly from shallow, weathered soil and regolith) that should be remobilized and transported by rivers within one to a few monsoons. Thus to the extent that ~50% and 90% of our RE catalogue had their largest or second largest landslides size at about $10^4$ m², we likely have short term sediment export on the same order than landslide rates. On millenial timescales, evacuation of sediments must depends on river transport capacity and remobilization of debris on hillslopes, likely linked to hydro-climatic forcings (Pratt-Sitaula et al., 2004, Cook et al., 2018). Recent modelling study suggest that fast (10-100 yr) evacuation of most of any large landslide deposit should be achievable due to river morphology self-adjustment (Croissant et al., 2017). However, the variable state of export of giant deposits ( >80% preserved for Latamrang and Dhumpu (5 kyr) deposits, but ~25% for the Braga (pre LGM) deposit, Weidinger, 2006), as well as evidence of substantial sediment storage in the high range (Pratt-Sitaula et al., 2004, Blothe and Korup, 2013, Stolle et al., 2018) suggest complex evacuation dynamics.  As a result, landslide erosion rates may be similar to or significantly larger than $^{10}$Be depending whether landslide evacuation over the last ~1kyr was efficient or not.  Nevertheless, the estimated total modern storage in the central Himalayas is ~100 km$^3$ within an area of >$10^5$ km$^2$ (Blothe and Korup, 2013), equivalent to a mean cover of 1m, or about 500 yr of landslide erosion, while fission track indicate that >1 mm/yr of erosion have been sustained for 10 Myr or more, clearly indicating that on million year time scales landslide deposit are effectively transported and storage is extremely minor."

555-560 : these comparisons between rates from different methods do not make much sense if you do not emphasis the context and particularities of these, as well as give more information about which data are used and their relevance to your area of interest. They seem to encompass very larges and diverse areas both along and across strike. In particular, to what extent are the lower bounds defined by data from the LH, which are actually outside of the investigated area?

>> For The fission track, we only consider the younger ages (0-4Ma) only in Central Nepal, and only in the Higher himalayas sequence (North and South section are similar in Thiede and Ehlers). Thiede and Ehlers give the mean modelled erosion best explaining the synthesis of cooling ages at 2.1 +/-0.5 mm/yr. For the rest of the range we use model for sections near the Western syntax, in the Sutlej, Central Nepal and Bhutan (Thiede and Ehlers 2013) and added the recent work of Abrahami and Wobus for Sikkim.

For the Sediment budget and 10Be we limit ourselve to small catchment as budget over catchment integrating over large spatial scales are euivalent to longer time sampling, ad ae generally larger and more steady (Lupker 2012, Morin 2015).

Specifically, Sediment budget in Nepal were only done in the High Himalayas, in several small tributaries of the Marsyangdi (Gabet2008). The others measurements are from Northwestern Himalayas and Pakistan near the syntax (Ali and Boers, 2007, Rao et al., Wulf et al., 2012). The 10 Be measurements are most widespread, from Sutlej (Scherler et al., 2014) to Bhutan (Portenga et al., 2015) with many measurements in central Nepal(Wobus et al., 2005, Godard et al., 2012, 2014, ) and some in Sikkim (Abrahami et al., 2016).

This is all presented and labelled in extended Fig 7.
Previous works have suggested that sub zones have different long term erosion (e.g., syntax are exhumed faster thant central Nepal…) and thus we display separately the fission track rates from different sub areas. For the short term measurements, given their variability, we show boxplot of all data, and of Nepal only. In both cases, in spite of uncertainties, a trend from short to long timescales is suggested by the available data.

: Portenga et al. (2005) actually in Bhutan.
>> We could not find any study by Portenga in 2005. If the referee means "Portenga 2015, in Bhutan" we do agree but find this reference useful because it presents a compilation from previous 10Be measurements across India, Nepal and Bhutan. We now state:  "(see compilation in Portenga et al., 2015)"

559-560 : given the very large reported ranges are these comparison really meaningful?
>> It is clear that we do not aim at reproducing the exact values but show that integrating landslide over the typical spatio-temporal scales sampled by 10Be yield a mean consistent with the one found by 10Be studies. Then if landslide are the dominant erosion process the random fluctuations around this mean should be very large for small area and timescales short compared to the recurrence of large landslides, as observed for 10Be and sediment budgets.

No changes made.

570-571 : bedrock landsliding will only be an efficient erosion mechanism if river incision is able to maintain local hillslope gradients close to the critical value, and mobilize the corresponding material over the timescale of interest.
>> Correct. We added: "Landslide dominant influence require the hillslopes to be coupled to rivers able to evacuate sediments and maintain steep slopes as it occur in the Himalayas"

576¢ a: "the observed increase in erosion rates from short to long timescales" this is very fuzzy to me, a plot of erosion rates vs integration time scale would probably help (with actual data, not just the ranges). See similar comments above.
>> We have made a new figure compiling fluvial sediment budget, 10Be denudation and fission tracks in Central Nepal as well as in other parts of the range. We now refer to it in the text. See comment earlier.

Figure 1 : I would expect this introductory situation figure to provide more context concerning the geology and climate of the area. Additional panels (same size and extent) with the corresponding information would be necessary, from my point of view, in particular if you want to support the hypothesis of homogeneity of many of these parameters made above.

>> We have added a Lithological map panel,with major faults, and possibly an ulift cros section as a color band ? Then another inset with the Rainfall map and glacierized areas.
Again we do not assume that all those parameters are equal, but that their net effects on landslide dynamics is not varying much. This does not say that these geographical variations may not matter for other processes, such as soil formation, underground water storage, river flow…

Figure 2 : add a vertical bar for the eq(s) date. Lot's of different symbol, some of which are defined in the legend text, not on the figure (orange square). Maybe use a matrix form for the legend (catchments as columns and type of inventories as rows)? Maybe add an upper panel with cumulative monsoon rainfall (same time axis)?

>> We now use a matrix form for the legend, with all symbols and earthquake lines

Figure 3 : recall the fitted parameters in the legend. A short title for each panel would help navigation (applicable for other figures)

>> I have added the power law exponent alpha and the term Beta controlling the exponential decay before the roll over in the figure caption.

Figure 4 : for the inset I would draw a vertical bar at 1 (and probably break the bins here)

>> We added the vertical line at 1 and inserted in the caption: "The black vertical line indicates correct prediction."
We prefer not to break the bin there, as centering the bin on 1 gives the number of monsoon correctly predicted within a certain factor of uncertainty.

Figure 5 : I am probably reading that wrong , but for the upper panel Y-axis label should not be "....landslide with size<A"?

>> The referee is correct and this typo has been corrected. We also added a gray zone to indicate that landslides < 1000m² is not modelled but neglected (as said in the text).

[revised manuscript text omitted]